# Distributionally Robust Ensemble of Lottery Tickets Towards Calibrated Sparse Network Training

**Hitesh Sapkota**    **Dingrong Wang**    **Zhiqiang Tao**    **Qi Yu** *
Rochester Institute of Technology
{hxs1943, dw7445, zhiqiang.tao, qi.yu}@rit.edu

## Abstract

The recently developed sparse network training methods, such as Lottery Ticket Hypothesis (LTH) and its variants, have shown impressive learning capacity by finding sparse sub-networks from a dense one. While these methods could largely sparsify deep networks, they generally focus more on realizing comparable accuracy to dense counterparts yet neglect network calibration. However, how to achieve calibrated network predictions lies at the core of improving model reliability, especially when it comes to addressing the overconfident issue and out-of-distribution cases. In this study, we propose a novel Distributionally Robust Optimization (DRO) framework to achieve an ensemble of lottery tickets towards calibrated network sparsification. Specifically, the proposed DRO ensemble aims to learn multiple diverse and complementary sparse sub-networks (tickets) with the guidance of uncertainty sets, which encourage tickets to gradually capture different data distributions from easy to hard and naturally complement each other. We theoretically justify the strong calibration performance by showing how the proposed robust training process guarantees to lower the confidence of incorrect predictions. Extensive experimental results on several benchmarks show that our proposed lottery ticket ensemble leads to a clear calibration improvement without sacrificing accuracy and burdening inference costs. Furthermore, experiments on OOD datasets demonstrate the robustness of our approach in the open-set environment.

## 1 Introduction

While there is remarkable progress in developing deep neural networks with densely connected layers, most of these dense networks have poor calibration performance [9], limiting their applicability in safety-critical domains like self-driving cars [4] and medical diagnosis [11]. The poor calibration is mainly due to the fact that there exists a good number of wrongly classified data samples (*i.e.,* low accuracy) with high confidence resulting from the memorization effect introduced by an over-parameterized architecture [27]. Recent sparse network training methods, such as Lottery Ticket Hypothesis (LTH) [6] and its variants [3, 2, 38, 18, 16, 35] generally assume that there exists a sparse sub-network (*i.e.*, lottery ticket) in a randomly initialized dense network, which could be trained in isolation and also match the performance of its dense counterpart network in terms of accuracy. While these methods may, to some extent, alleviate the overconfident issue, most of them require pre-training of a dense network followed by multi-step iterative pruning, making the overall training process highly costly, especially for large dense networks. Even for techniques that do not rely on pre-training and iterative pruning (*e.g.,* Edge Popup or EP [25]), their learning goal focuses on pushing the accuracy up to the original dense networks and hence may still exhibit a severely over-fitting behavior, leading to a poor calibration performance as demonstrated in Figure 1 (b).

Inspired by the recent success of using ensembles to estimate uncertainties [13, 34], a potential solution to realize well-calibrated predictions would be training multiple sparse sub-networks and

---

*Corresponding author

37th Conference on Neural Information Processing Systems (NeurIPS 2023).

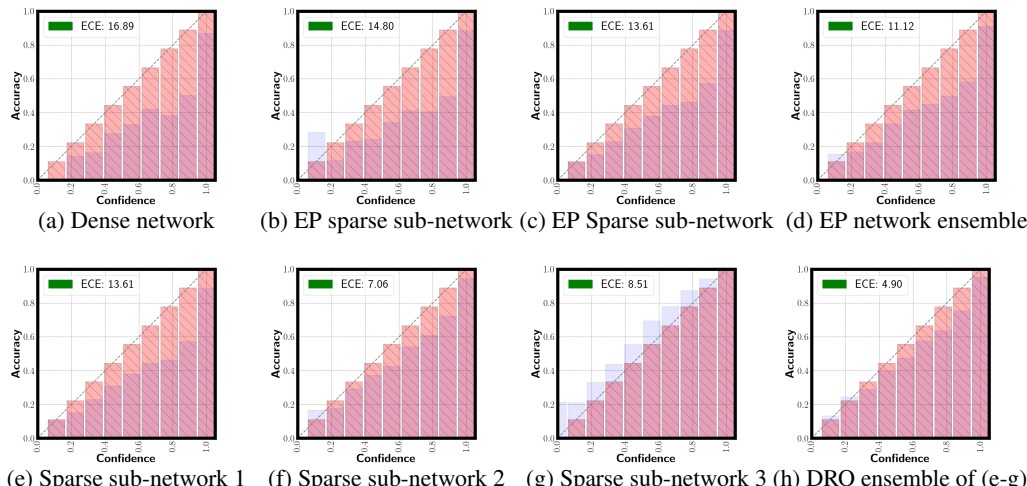

Figure 1: Calibration performance by expected calibration error (ECE) on Cifar100 dataset with ResNet101 architecture with density $\mathcal{K} = 15\%$. EP refers to the Edge Popup algorithm [25].

building an ensemble from them. As such, by leveraging accurate uncertainty quantification, the ensemble is expected to achieve better calibration. However, existing ensemble models of sparse networks rely on pre-training and iterative fine-tuning for learning each sub-network [18, 35], leading to a significant overhead for building the entire ensemble. Furthermore, an ensemble of independently trained sparse sub-networks does not necessarily improve the calibration performance. Since these networks are trained in a similar fashion from the same training data distribution, they could be strongly correlated such that the ensemble model will potentially inherit the overfitting behavior of each sub-network as shown in Figure 1(c). Therefore, the calibration capacity of sparse sub-network ensemble can be compromised as shown empirically in Figure 1 (d).

To further enhance the calibration of the ensemble, it is critical to ensure sufficient diversity among sparse sub-networks so that they are able to complement each other. One natural way to achieve diversity is to allow each sparse sub-network (ticket) to primarily focus on a specific part of training data distribution. This inspires us to leverage the AdaBoost [28] framework that sequentially finds tickets by manipulating training data distribution based on errors. By this means, the AdaBoost facilitates the training for a sequence of complementary sparse sub-networks. However, the empirical analysis (see Table 1) reveals that in the AdaBoost ensemble, most sub-networks (except for the first one) severely under-fit data leading to poor generalization ability. This is mainly because of the overfitting behavior of the first sub-network, which assigns very low training losses to the majority of data samples, making the subsequent sub-networks concentrate on very rare difficult samples that are likely to be outliers or noises. Hence, directly learning from these difficult samples without having global knowledge of the entire training distribution will result in the failure of subsequent training tickets and also hurt the overall calibration.

To this end, we need a more robust learning process for proper training of complementary sparse sub-networks, each of which can be learned in an efficient way to ensure the cost-effective construction of the entire ensemble. We propose a Distributionally Robust Optimization (DRO) framework to schedule learning an ensemble of lottery tickets (sparse sub-networks) with complimentary calibration behaviors that contribute to an overall well-calibrated ensemble as shown in Figure 1 (e-h). Our technique directly searches sparse sub-networks in a randomly initialized dense network without pre-training or iterative pruning. Unlike the AdaBoost ensemble, the proposed ensemble ticket method starts from the original training distribution and eventually allows learning each sub-network from different parts of the training distribution to enrich diversity. This is also fundamentally different from existing sparse ensemble models [18, 35], which attempt to obtain diverse sub-networks in a heuristic way by relying on different learning rates. As a result, these models offer no guaranteed complementary behavior among sparse sub-networks to cover a different part of training data, which is essential to alleviate the overfitting behavior of the learned sparse sub-networks. In contrast, we realize a principled scheduling process by changing the uncertainty set of DRO, where a small set pushes sub-networks learning with easy data samples and a large set focuses on the difficult ones (see Figure 2). By this means, the ticket ensemble governed by our DRO framework could work complementary and lead to much better calibration ability as demonstrated in Figure 1(h). On the one hand, we hypothesize that the ticket found with easy data samples will tend to be learned and

overfitted easily, resulting in overconfident predictions (Figure 1(e)). On the other hand, the ticket focused on more difficult data samples will be less likely to overfit and may become conservative and give under-confident predictions. Thus, it is natural to form an ensemble of such lottery tickets to complement each other in making calibrated predictions. As demonstrated in Figure 1 (h), owing to the diversity in the sparse sub-networks (e-g), the DRO ensemble exhibits better calibration ability. It is also worth noting that under the DRO framework, our sparse sub-networks already improve the calibration ability as shown in Figure 1 (f-g), which is further confirmed by our theoretical results.

Experiments conducted on three benchmark datasets demonstrate the effectiveness of our proposed technique compared to sparse counterparts and dense networks. Furthermore, we show through the experimentation that because of the better calibration, our model is being able to perform well on the distributionally shifted datasets [6] (CIFAR10-C and CIFAR100-C). The experiments also demonstrate that our proposed DRO ensemble framework can better detect open-set samples on varying confidence thresholds. The contribution of this work can be summarized as follows:

- a new sparse ensemble framework that combines multiple sparse sub-networks to achieve better calibration performance without dense network training and iterative pruning.
- a distributionally robust optimization framework that schedules the learning of an ensemble complementary sub-networks (tickets),
- theoretical justification of the strong calibration performance by showing how the proposed robust training process guarantees to lower the confidence of incorrect predictions in Theorem 2,
- extensive empirical evidence on the effectiveness of the proposed lottery ticket ensemble in terms of competitive classification accuracy and improved open-set detection performance.

## 2 Related Work

**Sparse networks training.** Sparse network training has received increasing attention in recent years. Representative techniques include lottery ticket hypothesis (LTH) [6] and its variants [5, 32]. To avoid training a dense network, supermasks have been used to find the winning ticket in the dense network without training network weights [38]. Edge-Popup (EP) extends this idea by leveraging training scores associated with the neural network weights and only weights with top scores are used for predictions. There are two key limitations to most existing LTH techniques. First, most of them require pre-training of a dense network followed by multi-step iterative pruning making the overall training process expensive. Second, their learning objective remains as improving the accuracy up to the original dense networks and may still suffer from over-fitting (as shown in Figure 1).

**Sparse network ensemble.** There are recent advancements in building ensembles from sparse networks. A pruning and regrowing strategy has been developed in a model, called CigL [16], where dropout serves as an implicit ensemble to improve the calibration performance. CigL requires weight updates and performs pruning and growing for multiple rounds, leading to a high training cost. Additionally, dropping many weights may lead to a performance decrease, which prevents building highly sparse networks. This idea has been further extended by using different learning rates to generate different typologies of the network structure for each sparse network [18, 35]. While diversity among sparse networks can be achieved, there is no guarantee that this can improve the calibration performance of the final ensemble. In fact, different networks may still learn from the training data in a similar way. Hence, the learned networks may exhibit similar overfitting behavior with a high correlation, making it difficult to generate a well-calibrated ensemble. In contrast, the proposed DRO ensemble schedules different sparse networks to learn from complementary parts of the training distribution, leading to improved calibration with theoretical guarantees.

**Model calibration.** Various attempts have been proposed to make the deep models more reliable either through calibration [9, 24, 32] or uncertainty quantification [7, 30]. Post-calibration techniques have been commonly used, including temperature scaling [24, 9], using regularization to penalize overconfident predictions [23]. Recent studies show that post-hoc calibration falls short of providing reliable predictions [22]. Most existing techniques require additional post-processing steps and an additional validation dataset. In our setting, we aim to improve the calibration ability of sparse networks without introducing additional post-calibration steps or validation dataset.

## 3 Methodology

Let $\mathcal{D}_{\mathcal{N}} = \{\mathbf{X}, \mathbf{Y}\} = \{(\mathbf{x}_1, y_1), .., (\mathbf{x}_N, y_N)\}$ be a set of training samples where each $\mathbf{x}_n \in \mathbb{R}^D$ is a D-dimensional feature vector and $y_n \in [1, C]$ be associated label with $C$ total classes. Let $M$ be

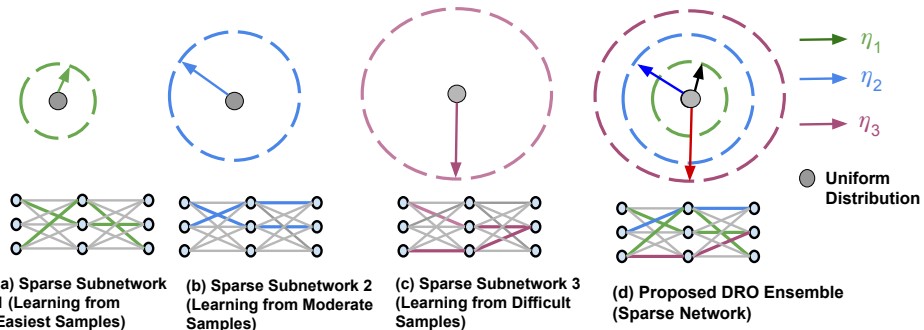

Figure 2: Robust ensemble where $\eta$ defines the size of an uncertainty set with $\eta_1 \leq \eta_2 \leq \eta_3$.

the total number of base learners used in the given ensemble technique. Further, consider $\mathcal{K}$ to be the density ratio in the given network, which denotes the percentage of weights we keep during the training process. The major notations are summarized in the Appendix.

### 3.1 Preliminaries

**Edge-Popup (EP)** [25]. EP finds a lottery ticket (sparse sub-network) from a randomly initialized dense network based on the score values learned from training data. Specifically, to find the sub-network with density $\mathcal{K}$, the algorithm optimizes the scores associated with each weight in the dense network. During the forward pass, the top-$\mathcal{K}$ weights in each layer are selected based on their scores. During the backward pass, scores associated with all weights are updated, which allows potentially useful weights that are ignored in previous forward passes to be re-considered.

**Expected calibration error.** Expected Calibration Error (ECE) measures the correspondence between predicted probability and empirical accuracy [20]. Specifically, mis-calibration is computed based on the difference in expectation between confidence and accuracy: $\mathbb{E}_{\hat{p}}\left[|\mathbb{P}(\hat{y} = y|\hat{p} = p) - p|\right]$. In practice, we approximate the expectation by partitioning confidences into $T$ bins (equally spaced) and take the weighted average on the absolute difference between each bins' accuracy and confidence. Let $B_t$ denote the $t$-th beam and we have ECE $= \sum_{t=1}^{T} \frac{|B_t|}{N} |acc(B_t) - conf(B_t)|$.

### 3.2 Distributionally Robust Ensemble (DRE)

As motivated in the introduction, to further enhance the calibration of a deep ensemble, it is instrumental to introduce sufficient diversity among the component sparse sub-networks so that they can complement each other when forming the ensemble. One way to achieve diversity is to allow each sparse sub-network to primarily focus on a specific part of the training data distribution. Figure 2 provides an illustration of this idea, where the training data can be imagined to follow a multivariate Gaussian distribution with the red dot representing its mean. In this case, the first sub-network will learn the most common patterns by focusing on the training data close to the mean. The subsequent sub-networks will then learn relatively rare patterns by focusing on other parts of the training data (*e.g.,* two or three standard deviations from the mean).

**AdaBoost ensemble.** The above idea inspires us to leverage the AdaBoost framework [28] to manipulate the training distribution that allows us to train a sequence of complementary sparse sub-networks. In particular, we train the first sparse sub-network from the original training distribution, where each data sample has an equal probability to be sampled. In this way, the first sparse sub-network can learn the common patterns from the most representative training samples. Starting from the second sub-network, the training distribution is changed according to the losses suffered from the previous sub-network during the last round of training. This allows the later sub-networks to focus on the difficult data samples by following the spirit of AdaBoost.

However, our empirical results reveal that in the AdaBoost ensemble, most sub-networks (except for the first one) severely underfit the training data, leading to a rather poor generalization capability. This is caused by the overfitting behavior of the first sparse sub-network, which assigns very small training losses to a majority of data samples. As a result, the subsequent sub-networks can only focus on a limited number of training samples that correspond to relatively rare patterns (or even outliers and noises) in the training data. Directly learning from these difficult data samples without a general knowledge of the entire training distribution will result in the failure of training the sub-networks.

**Distributionally robust ensemble (DRE).** To tackle the challenge as outlined above, we need a more robust learning process to ensure proper training of complementary sparse sub-networks. Different from the AdaBoost ensemble, the training of all sub-networks starts from the original training distribution in the DRO framework. Meanwhile, it also allows each sub-network to eventually focus on learning from different parts of the training distribution to ensure the desired diverse and complementary behavior. Let $l(\mathbf{x}_n, \Theta)$ denote the loss associated with the $n^{th}$ data sample with $\Theta$ being the parameters in the sparse sub-network. Then, the total loss is given by

$$\mathcal{L}^{\text{Robust}}(\Theta) = \max_{\mathbf{z} \in \mathcal{U}^{\text{Robust}}} \sum_{n=1}^{N} z_n l(\mathbf{x}_n, \Theta) \tag{1}$$

The uncertainty set defined to assign weights $\mathbf{z}$ is given as

$$\mathcal{U}^{\text{Robust}} := \left\{ \mathbf{z} \in \mathbb{R}^N : \mathbf{z}^\top \mathbf{1} = 1, \mathbf{z} \geq 0, D_f(\mathbf{z} \| \frac{\mathbf{1}}{N}) \leq \eta \right\} \tag{2}$$

where $D_f(\mathbf{z} \| \mathbf{q})$ is $f$-divergence between two distributions $\mathbf{z}$ and $\mathbf{q}$ and $\eta$ controls the size of the uncertainty set and $\mathbf{1} \in 1^N$ is $N$-dimensional unit vector. Depending on the $\eta$ value, the above robust framework instantiates different sub-networks. For example, by making $\eta \to \infty$, we have $\mathcal{U}^{\text{Robust}} = \left\{ \mathbf{z} \in \mathbb{R}^N : \mathbf{z}^\top \mathbf{1} = 1, \mathbf{z} \geq 0, D_f(\mathbf{z} \| \frac{\mathbf{1}}{N}) \leq \infty \right\}$. In this case, we train a sub-network by only using the most difficult sample in the training set. On the other extreme with $\eta \to 0$, we have $\mathcal{U}^{\text{Robust}} = \left\{ \mathbf{z} \in \mathbb{R}^N : \mathbf{z}^\top \mathbf{1} = 1, \mathbf{z} \geq 0, D_f(\mathbf{z} \| \frac{\mathbf{1}}{N}) \leq 0 \right\}$, which assigns equal weights to all data samples. So, the sub-network learns from the original training distribution.

To fully leverage the key properties of the robust loss function as described above, we propose to perform distributionally robust ensembling learning to generate a diverse set of sparse sub-networks with well-controlled overfitting behavior that can collectively achieve superior calibration performance. The training process starts with a relatively small $\eta$ value to ensure that the initially generated sub-networks can adequately capture the general patterns from the most representative data samples in the original training distribution. The training proceeds by gradually increasing the $\eta$ value, which allows the subsequent sub-networks to focus on relatively rare and more difficult data samples. As a result, the later generated sub-networks tend to produce less confident predictions that complement the sub-networks generated in the earlier phase of the training process. This diverse and complementary behavior among different sparse sub-networks is clearly illustrated in Figure 1 (e)-(g). During the ensemble phase, we combine the predictions of different sub-networks in the logit space by taking the mean and then performing the softmax. In this way, the sparse sub-networks with high $\eta$ values help to lower the overall confidence score, especially those wrongly predicted data samples. Furthermore, the sub-networks with lower $\eta$ values help to bring up the confidence score of correctly predicted data samples. Thus, the overall confidence score will be well compensated, resulting in a better calibrated ensemble.

### 3.3 Theoretical Analysis

In this section, we theoretically justify why the proposed DRE framework improves the calibration performance by extending the recently developed theoretical framework on multi-view learning [1]. In particular, we will show how it can effectively lower the model's false confidence on its wrong predictions resulting from spurious correlations. To this end, we first define the problem setup that includes some key concepts used in our theoretical analysis. We then formally show that DRO helps to tackle the spurious correlations by learning from less frequent features that characterize difficult data samples in a training dataset. This important property further guarantees better calibration performance of DRO as we show in the main theorem. It is worth to note that our theoretical analysis is primarily from the spurious correlation perspective. This is only one of the potential sources that can lead to over-confidence, resulting in poor-calibration in neural networks. The goal is offer deeper insights on why the proposed approach is able to improve the calibration performance resulting from spurious correlations by effectively lowering the model's false confidence on its wrong predictions.

**Problem setup.** Assume that each data sample $\mathbf{x}_n \in \mathbb{R}^D$ is divided into $P$ total patches, where each patch is a $d$-dimensional vector. For the sake of simplicity, let us assume each class $c \in [1, C]$ has two characterizing (major) features $\mathbf{v}_c = \{\mathbf{v}_{c,l}\}_{l=1}^L$ with $L = 2$. For example, the features for `Cars` could be `Headlights` and `Tires`. Let $\mathcal{D}_N^S$ and $\mathcal{D}_N^M$ denote the set of *single-view* and *multi-view* data

samples, respectively, which are formally defined as

$$\begin{cases} \{\mathbf{x}_n, y_n\} \in \mathcal{D}_N^S \text{ if one of } \mathbf{v}_{c,1} \text{ or } \mathbf{v}_{c,2} \text{ appears along with some noise features} \\ \{\mathbf{x}_n, y_n\} \in \mathcal{D}_N^M \text{ if both } \mathbf{v}_{c,1} \text{ and } \mathbf{v}_{c,2} \text{ appears along with some noise features} \end{cases} \quad (3)$$

The noise features (also called minor features) refer to those that do not characterize (or differentiate) a given class $c$ (*e.g.,* being part of the background). In important applications like computer vision, images supporting such a "multi-view" structure is very common [1]. For example, for most car images, we can observe all main features, such as `Wheels`, `Tires`, and `Headlights` so they belong to $\mathcal{D}_N^M$. Meanwhile, there may also be images, where multiple features are missing. For example, if the car image is taken from the front, the tire and wheel features may not be captured. In most real-world datasets, such single-view data samples are usually much limited as compared to their multi-view counterparts. The Appendix provides concrete examples of both single and multi-view images. Let us consider $(\mathbf{x}, y) \in \mathcal{D}_N^S$ with the major feature $\mathbf{v}_{c,l}$ where $y = c$. Then each patch $\mathbf{x}^p \in \mathbb{R}^d$ can be expressed as

$$\mathbf{x}^p = a^p \mathbf{v}_{c,l} + \sum_{\mathbf{v}' \in \cup \setminus \mathbf{v}_c} \alpha^{p, \mathbf{v}'} \mathbf{v}' + \epsilon^p \quad (4)$$

where $\cup = \{\mathbf{v}_{c,1}, \mathbf{v}_{c,2}\}_{c=1}^C$ is collection of all features, $a^p > 0$ is the weight allocated to feature $\mathbf{v}_{c,l}$, $\alpha^{p, \mathbf{v}'} \in [0, \gamma]$ is the weight allocated to the noisy feature $\mathbf{v}'$ that is not present in feature set $\mathbf{v}_c$ *i.e.,* $\mathbf{v}' \in \cup \setminus \mathbf{v}_c$, and $\epsilon^p \sim \mathcal{N}(0, (\sigma^p)^2 \mathbb{1})$ is a random Gaussian noise. In (4), a patch $\mathbf{x}^p$ in a single-view sample $\mathbf{x}$ also contains set of minor (noise) features presented from other classes *i.e.,* $\mathbf{v}' \in \cup \setminus \mathbf{v}_c$ in addition to the main feature $\mathbf{v}_{c,l}$. Since $\mathbf{v}_{c,l}$ characterizes class $c$, we have $a^p > \alpha^{p, \mathbf{v}'}; \forall \mathbf{v}' \in \cup \setminus \mathbf{v}_c$. However, since the single-view data samples are usually sparse in the training data, it may prevent the model from accumulating a large $a^p$ for $\mathbf{v}_{c,l}$ as shown Lemma 1 below. In contrast, some noise $\mathbf{v}'$ may be selected as the dominant feature (due to spurious correlations) to minimize the errors of specific training samples, leading to potential overfitting of the model.

We further assume that the network contains $H$ convolutional layers, which outputs $F(\mathbf{x}; \Theta) = (F_1(\mathbf{x}), ... F_C(\mathbf{x})) \in \mathbb{R}^C$. The logistic output for the $c^{th}$ class can be represented as

$$F_c(\mathbf{x}) = \sum_{h \in [H]} \sum_{p \in [P]} \text{ReLU}[\langle \Theta_{c,h}, \mathbf{x}^p \rangle] \quad (5)$$

where $\Theta_{c,h}$ denote the $h^{th}$ convolution layer (feature map) associated with class $c$. Under the above data and network setting, we propose the following lemma.

**Lemma 1.** *Let $\mathbf{v}_{c,l}$ be the main feature vector present in the single-view data $\mathcal{D}_N^S$. Assume that number of single-view data samples containing feature $\mathbf{v}_{c,l}$ is limited as compared with the rest,* i.e., *$N_{\mathbf{v}_{c,l}} \ll N_{\cup \setminus \mathbf{v}_{c,l}}$. Then, at any iteration $t > 0$, we have*

$$\langle \Theta_{c,h}^{t+1}, \mathbf{v}_{c,l} \rangle = \langle \Theta_{c,h}^t, \mathbf{v}_{c,l} \rangle + \beta \max_{\mathbf{z} \in \mathcal{U}} \sum_{n=1}^N z_n \left[ \mathbb{1}_{y_j = c} (V_{c,h,l}(\mathbf{x}_n) + \kappa)(1 - \text{SOFT}_c(F(\mathbf{x}_n))) \right] \quad (6)$$

*where $\kappa$ is a dataset specific constant, $\beta$ is the learning rate, $\text{SOFT}_c$ is the softmax output for class $c$, and $V_{c,h,l}(\mathbf{x}_j) = \sum_{p \in \mathcal{P}_{\mathbf{v}_{c,l}}(\mathbf{x}_j)} \text{ReLU}(\langle \Theta_{c,h}, \mathbf{x}_j^p \rangle a^p)$ with $\mathcal{P}_{\mathbf{v}_{c,l}}(\mathbf{x}_j)$ being the collection of patches containing feature $\mathbf{v}_{c,l}$ in $\mathbf{x}_j$. The set $\mathcal{U}$ is an uncertainty set that assigns a weight to each data sample based on it loss. In particular, the uncertainty set under DRO is given as in (2) and we further define the uncertainty set under ERM: $\mathcal{U}^{ERM} := \{\mathbf{z} \in \mathbb{R}^N : z_n = \frac{1}{N}; \forall n \in [1, N]\}$. Learning via the robust loss in (1) leads to a stronger correlation between the network weights $\Theta_{c,h}$ and the single-view data feature $\mathbf{v}_{c,l}$:*

$$\{\langle \Theta_{c,h}^t, \mathbf{v}_{c,l} \rangle\}_{Robust} > \{\langle \Theta_{c,h}^t, \mathbf{v}_{c,l} \rangle\}_{ERM}; \forall t > 0 \quad (7)$$

**Remark.** The robust loss $\mathcal{L}^{\text{Robust}}$ forces the model to learn from the single-view samples (according to the loss) by assigning a higher weight. As a result, the network weights will be adjusted to increase the correlation with the single-view data features $\mathbf{v}_{c,l}$ due to Lemma 1. In contrast, for standard ERM, weight is uniformly assigned to all samples. Due to the sparse single-view data features (which also makes them more difficult to learn from, leading to a larger loss), the model does not grow sufficient

correlation with $\mathbf{v}_{c,l}$. In this case, the ERM model instead learns to memorize some noisy feature $\mathbf{v}'$ introduced through certain spurious correlations. For a testing data sample, the ERM model may confidently assign it to an incorrect class $k$ according to the noise feature $\mathbf{v}'$. In the theorem below, we show how the robust training proces can effectively lower the confidence of incorrect predictions, leading to an improved calibration performance.

**Theorem 2.** *Given a new testing sample* $\mathbf{x} \in \mathcal{D}_S^N$ *containing* $\mathbf{v}_{c,l}$ *as the main feature and a dominant noise feature* $\mathbf{v}'$ *that is learned due to memorization, we have*

$$\{SOFT_k(\mathbf{x})\}_{Robust} < \{SOFT_k(\mathbf{x})\}_{ERM} \tag{8}$$

*where* $\mathbf{v}'$ *is assumed to be a main feature characterizing class* $k$.

**Remark.** For ERM, due to the impact of the dominate noisy feature $\mathbf{v}'$, it assigns a large probability to class $k$ since $\mathbf{v}'$ is one of its major features, leading to high confidence for an incorrect prediction. In contrast, the robust learning process allows the model to learn a stronger correlation with the main feature $\mathbf{v}_{c,l}$ as shown in Lemma 1. Thus, the model is less impacted by the noise feature $\mathbf{v}'$, resulting in reduced confidence in predicting the wrong class $k$. Such a key property guarantees an improved calibration performance, which is clearly verified by our empirical evaluation. It is also worth noting that Theorem 2 does not necessarily lead to better classification accuracy. This is because (8) only ensures that that the false confidence is lower than an ERM model, but there is no guarantee that $\{\text{SOFT}_k(\mathbf{x})\}_{Robust} < \{\text{SOFT}_c(\mathbf{x})\}_{Robust}$. It should be noted that our DRE framework ensures diverse sparse sub-network focusing on different single-view data samples from different classes. As such, an ensemble of those diverse sparse subnetworks provides maximum coverage of all features (even the weaker one) and therefore can ultimately improve the calibration performance. The detailed proofs are provided in the Appendix.

## 4 Experiments

We perform extensive experimentation to evaluate the distributionally robust ensemble of sparse sub-networks. Specifically, we test the ability of our proposed technique in terms of calibration and classification accuracy. For this, we consider three settings: (a) general classification, (b) out-of-distribution setting where we have in-domain data but with different distributions, and (c) open-set detection, where we have unknown samples from new domains.

### 4.1 Experimental Settings

**Dataset description.** For the general classification setting, we consider three real-world datasets: Cifar10, Cifar100 [12], and TinyImageNet [14]. For the out-of-distribution setting, we consider the corrupted version of the Cifar10 and Cifar100 datasets which are named Cifar10-C and Cifar100-C [10]. It should be noted that in this setting, we train all models in clean dataset and perform testing in the corrupted datasets. For open-set detection, we use the SVHN dataset [21] as the open-set dataset and Cifar10 and Cifar100 as the close-set data. A more detailed description of each dataset is presented in the Appendix.

**Evaluation metrics.** To assess the model performance in the first two settings, we report the classification accuracy ($\mathcal{ACC}$) along with the Expected Calibration Error ($\mathcal{ECE}$). In the case of open-set detection, we report open-set detection for different confidence thresholds.

**Implementation details.** In all experiments, we use a family of ResNet architectures with two density levels: 9% and 15%. To construct an ensemble, we learn 3 sparse sub-networks each with a density of 3% for the total of 9% density and that of 5% density for the total of density 15%. All experiments are conducted with the 200 total epochs with an initial learning rate of 0.1 and a cosine scheduler function to decay the learning rate over time. The last-epoch model is taken for all analyses. For the training loss, we use the EP-loss in our DRO ensemble that optimizes the scores for each weight and finally selects the sub-network from the initialized dense network for the final prediction. The selection is performed based on the optimized scores. More detailed information about the training process and hyperparameter settings can be found in the Appendix.

### 4.2 Performance Comparison

In our comparison study, we include baselines that are relevant to our technique and therefore we primarily focus on the LTH-based techniques. Specifically, we include the initial lottery ticket hypothesis (LTH) [6] that iteratively performs pruning from a dense network until the randomly

Table 1: Accuracy and ECE performance with 9% density for Cifar10 and Cifar100.

| Training Type | Approach | Cifar10 | | | | Cifar100 | | | |
|---|---|---|---|---|---|---|---|---|---|
| | | ResNet50 | | ResNet101 | | ResNet101 | | ResNet152 | |
| | | $ACC$ | $ECE$ | $ACC$ | $ECE$ | $ACC$ | $ECE$ | $ACC$ | $ECE$ |
| | Dense$^\dagger$ | 94.82 | 5.87 | 95.12 | 5.99 | 76.40 | 16.89 | 77.97 | 16.73 |
| Dense Pre-training | L1 Pruning [17] | 93.45 | 5.31 | 93.67 | 6.14 | 75.11 | 15.89 | 75.12 | 16.24 |
| | LTH [6] | 92.65 | 3.68 | 92.87 | 6.02 | 74.09 | 15.45 | 74.41 | 16.12 |
| | DLTH [3] | 93.27 | 5.87 | 95.12 | 7.09 | 77.29 | 16.64 | 77.86 | 17.26 |
| | Mixup [32] | 92.86 | 3.68 | 93.06 | 6.01 | 74.15 | 15.41 | 74.28 | 16.05 |
| Sparse Training | CigL [16] | 92.39 | 5.06 | 93.41 | 4.60 | 76.40 | 9.30 | 76.46 | 9.91 |
| | DST Ensemble [18] | 88.87 | 2.02 | 84.93 | 0.8 | 63.57 | 7.23 | 63.22 | 6.18 |
| | Sup-ticket [35] | 94.52 | 3.30 | 95.04 | 3.10 | 78.28 | 10.20 | 78.60 | 10.50 |
| Mask Training | AdaBoost | 93.12 | 5.13 | 94.15 | 5.46 | 75.15 | 22.96 | 75.89 | 24.54 |
| | EP [25] | 94.20 | 3.97 | 94.35 | 4.03 | 75.05 | 14.62 | 75.68 | 14.41 |
| | SNE | 94.70 | 2.51 | 94.48 | 3.51 | 75.69 | 9.02 | 75.22 | 10.89 |
| | **DRE (Ours)** | 94.60 | **0.7** | 94.28 | **0.7** | 74.68 | **1.20** | 74.37 | **2.09** |

Table 2: Accuracy and ECE on TinyImageNet.

| Training Type | Approach | $\mathcal{K} = 9\%$ | | | | $\mathcal{K} = 15\%$ | | | |
|---|---|---|---|---|---|---|---|---|---|
| | | ResNet101 | | WideResNet101 | | ResNet101 | | WideResNet101 | |
| | | $ACC$ | $ECE$ | $ACC$ | $ECE$ | $ACC$ | $ECE$ | $ACC$ | $ECE$ |
| | Dense$^\dagger$ | 71.28 | 15.58 | 72.57 | 16.96 | 71.28 | 15.58 | 72.57 | 16.96 |
| Dense Pre-training | L1 Pruning [17] | 68.85 | 14.72 | 69.78 | 16.38 | 70.24 | 14.24 | 70.98 | 15.36 |
| | LTH [6] | 69.23 | 13.97 | 69.13 | 15.34 | 70.16 | 13.63 | 70.25 | 14.24 |
| | DLTH [3] | 70.12 | 16.15 | 71.36 | 18.35 | 71.68 | 15.88 | 72.97 | 17.21 |
| | Mixup [32] | 69.34 | 14.24 | 69.25 | 15.59 | 70.28 | 14.31 | 70.39 | 14.57 |
| Mask Training | AdaBoost | 69.52 | 17.23 | 68.66 | 19.46 | 70.12 | 16.57 | 70.24 | 18.35 |
| | EP [25] | 69.88 | 10.78 | 71.57 | 9.82 | 70.46 | 11.99 | 70.71 | 12.41 |
| | SNE | 71.28 | 4.64 | 73.32 | 5.48 | 72.20 | 6.57 | 74.56 | 6.55 |
| | **DRE (Ours)** | 71.68 | **3.48** | 74.04 | **2.82** | 72.00 | **1.52** | 73.72 | **1.08** |

Table 3: Accuracy and ECE performance on out-of-distribution datasets.

| Training Type | Approach | Cifar10 | | | | Cifar100 | | | |
|---|---|---|---|---|---|---|---|---|---|
| | | ResNet50 | | ResNet101 | | ResNet101 | | ResNet152 | |
| | | $ACC$ | $ECE$ | $ACC$ | $ECE$ | $ACC$ | $ECE$ | $ACC$ | $ECE$ |
| | Dense$^\dagger$ | 79.65 | 19.63 | 79.65 | 19.63 | 54.75 | 35.32 | 54.75 | 35.32 |
| Dense Pre-training | L1 Pruning [17] | 77.34 | 17.95 | 76.39 | 17.89 | 52.06 | 31.45 | 51.67 | 30.98 |
| | LTH [6] | 75.85 | 17.88 | 76.15 | 17.62 | 50.79 | 31.23 | 51.35 | 30.56 |
| | DLTH [3] | 79.67 | 21.74 | 80.12 | 20.31 | 54.82 | 37.55 | 55.12 | 35.74 |
| | Mixup [32] | 76.35 | 17.74 | 76.88 | 17.55 | 51.36 | 31.12 | 51.92 | 30.35 |
| Sparse Training | CigL [16] | 70.80 | 21.04 | 69.84 | 21.42 | 49.42 | 25.86 | 51.49 | 24.13 |
| | Sup-ticket [35] | 72.89 | 17.80 | 73.01 | 18.82 | 48.80 | 24.99 | 48.81 | 25.62 |
| Mask Training | AdaBoost | 75.94 | 22.96 | 74.55 | 21.46 | 51.36 | 38.45 | 51.25 | 38.34 |
| | EP [25] | 77.58 | 17.82 | 77.73 | 17.46 | 52.18 | 30.60 | 52.14 | 29.48 |
| | SNE | 78.93 | 15.73 | 78.61 | 15.56 | 54.74 | 24.22 | 54.00 | 20.54 |
| | **DRE (Ours)** | 78.57 | **10.92** | 78.00 | **10.19** | 54.11 | **14.28** | 53.21 | **8.13** |

initialized sub-network with a given density is reached. Once the sub-network is found, the model trains the sub-network using the training dataset. Similarly, we also include L1 pruning [17]. We also include three approaches CigL [16], Sup-ticket [35], DST Ensemble [18] which are based on the pruning and regrowing sparse network training strategies. From Venkatesh et al. [32] we consider MixUp strategy as a comparison baseline as it does not require multi-step forward passes. A dense network is also included as a reference (denoted as Dense$^\dagger$). Furthermore, we report the performance obtained using the EP algorithm [25] on a single model with a given density. Finally, we also include the deep ensemble technique (*i.e.,* Sparse Network Ensemble (SNE), where each base model is randomly initialized and independently trained. The approaches that require pre-training of a dense network are categorized under the *Dense Pre-training* category. Those performing sparse network training but actually updating the network parameters are grouped as *Sparse Training*. It should be noted that sparse training techniques still require iterative pruning and regrowing. Finally, techniques that attempt to search the best initialized sparse sub-network through mask update (*e.g.,* EP) are grouped as *Mask Training*.

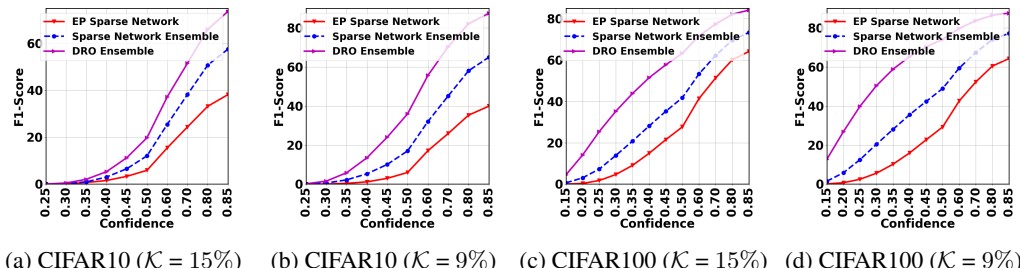

|                    |                    |                      |                     |
|:------------------:|:------------------:|:--------------------:|:-------------------:|
| (a) CIFAR10 ($\mathcal{K} = 15\%$) | (b) CIFAR10 ($\mathcal{K} = 9\%$) | (c) CIFAR100 ($\mathcal{K} = 15\%$) | (d) CIFAR100 ($\mathcal{K} = 9\%$) |

Figure 3: Open-set detection performance on different confidence thresholds.

**General classification setting.** In this setting, we consider clean Cifar10, Cifar100, and TinyImageNet datasets. Tables 1, 2, and 6 (in the Appendix) show the accuracy and calibration error for different models with density $9\%$ and $15\%$. It should be noted that for the TinyImageNet dataset, we could not run the Sparse Training techniques due to the computation issue (*i.e.,* memory overflow). This may be because sparse training techniques require maintaining additional parameters for the pruning and regrowing strategy. In the Appendix, we have made a comparison of the proposed DRE with those baselines on a lower architecture size. There are three key observations we can infer from the experimental results. First, sparse networks are able to maintain or improve the generalization performance (in terms of accuracy) with better calibration, which can be seen by comparing dense network performance with the edge-popup algorithm. Second, the ensemble in general helps to further lower the calibration error (lower the better). For example, in all datasets, standard ensemble (SNE) consistently improves the EP model. Finally, the proposed DRE significantly improves the calibration performance by diversifying base learners and allow each sparse sub-network to focus on different parts of the training data. The strong calibration performance provides clear empirical evidence to justify our theoretical results.

**Out-of-distribution classification setting.** In this setting, we assess the effectiveness of the proposed techniques on out-of-distribution samples. Specifically, [10] provide the Cifar10-C and Cifar100-C validation datasets which are different than that of the original clean datasets. They apply different corruptions (such as blurring noise, and compression) to shift the distribution of the datasets. We assess those corrupted datasets using the models trained using the clean dataset. Table 3 shows the performance using different architectures. In this setting, we have not included DST Ensemble, because: (a) its accuracy is far below the SOTA performance, and (b) same training mechanism as that of the Sup-ticket, whose performance is reported. As shown, the proposed DRE provides much better calibration performance even with the out of distribution datasets.

**Open-set detection setting.** In this setting, we demonstrate the ability of our proposed DRO ensemble in detecting open-set samples. For this, we use the SVHN dataset as an open-set dataset. Specifically, if we have a better calibration, we would be able to better differentiate the open-set samples based on the confidence threshold. For this, we randomly consider $20\%$ of the total testing in-distribution dataset as the open-set samples from the SVHN dataset. The reason for only choosing a subset of the dataset is to imitate the practical scenario where we have very few open-set samples compared to the close-set samples. We treat the open-set samples as the positive and in-distribution (close-set) ones as the negative. Since this is a binary detection problem, we compute the F-score [8] at various thresholds, which considers both precision and recall. Figure 3 shows the performance for the proposed technique along with comparative baselines. As shown, our proposed DRE (refereed as DRO Ensemble) always stays on the top for various confidence thresholds which demonstrates that strong calibration performance can benefit DRE for open-set detection as compared to other baselines.

### 4.3 Ablation Study

In this section, we investigate the impact of the backbone architecture along with the size of the ensemble. Additional ablation studies are presented in Appendix D.9.

**Performance analysis of different backbones.** Table 4 (a) reports the performance of Cifar10 from both DRE and EP using different backbone architectures. In case of WideResNet28-10, the calibration error is low without sacrificing the accuracy. It also demonstrates that the superior performance of DRE is not limited to a specific backbone. In case of ViT, DRE still achieves a much lower calibration error than EP. However, using ViT as a backbone, the accuracy from both EP and DRE is lower and ECE is higher than other backbones. Existing studies show that without pretraining, the lack of useful

Table 4: ACC and ECE with different backbones and number of subnetworks

| Approach | WideResNet28-10 | | ViT | | | Approach | ResNet101 | | ResNet152 | |
|---|---|---|---|---|---|---|---|---|---|---|
| | $\mathcal{ACC}$ | $\mathcal{ECE}$ | $\mathcal{ACC}$ | $\mathcal{ECE}$ | | | $\mathcal{ACC}$ | $\mathcal{ECE}$ | $\mathcal{ACC}$ | $\mathcal{ECE}$ |
| EP | 94.12 | 4.53 | 86.16 | 10.01 | | DRE (M = 3) | 94.87 | 1.71 | 94.74 | 1.34 |
| DRE | 93.98 | **1.93** | 85.53 | **4.18** | | DRE (M = 5) | 94.79 | 0.84 | 94.69 | 0.62 |

(a) Different backbones on Cifar10 Dataset.      (b) Different $M$ values on Cifar10 with $\mathcal{K} = 15\%$.

inductive biases for ViT can cause performance drop [1]. Since no pretraining is conducted in both EP and DRE, it causes a lower accuracy (and a higher ECE).

**Impact of number of sparse-sub-networks.** In this analysis, we study the impact of number of sparse sub-networks. It should be noted that our work is not limited only for $M = 3$. We can instead increase the $M$ value. For example, Table 4 (b) shows the performance for an ensemble model with $M = 5$, where each sub-network is trained with $\mathcal{K} = 3\%$, leading to a total $\mathcal{K} = 15\%$. We also show the performance with $M = 3$, where each sub-network is trained with $\mathcal{K} = 5\%$. As can be seen, if there is a sufficient learning capacity for each sub-network, the ECE score can further improve with the increase of $M$.

### 4.4 Qualitative Analysis

In this section, we provide illustrative examples to further justify why the proposed DRE is better calibrated as compared to existing baselines. Figure 4 (a)-(d) show the confidence values for the wrongly classified samples by different sparse sub-networks in the DRE ensemble. As can be seen, each sparse sub-network provides confidence values in different ranges, where the sub-network in (a) is learned from most representative samples and the one in (c) is from the most difficult ones. As these sub-networks are complementary with each other, the DRE has a much better confidence distribution for incorrect samples than the baselines. In contrast, as shown in Figure 8 of Appendix D.12, all the baselines, including the dense network, EP sparse networks, and the SNE, tend to allocate much higher confidence values to wrongly classified examples, leading to poor calibration. In case of correctly classified samples, the proposed DRE generates confident predictions and thereby not compromising the calibration performance, which is further discussed in Appendix D.12.

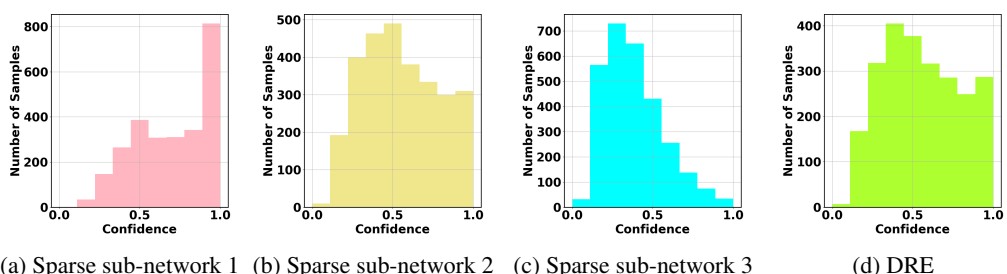

(a) Sparse sub-network 1  (b) Sparse sub-network 2  (c) Sparse sub-network 3      (d) DRE

Figure 4: Confidence scores of incorrectly classified samples in CIFAR100 with ResNet101.

## 5   Conclusion

In this paper, we proposed a novel DRO framework, called DRE, that achieves an ensemble of lottery tickets towards calibrated network sparsification. Specifically, with the guidance of uncertainty sets under the DRO framework, the proposed DRE aims to learn multiple diverse and complementary sparse sub-networks (tickets) where uncertainty sets encourage tickets to gradually capture different data distributions from easy to hard and naturally complement each other. We have theoretically justified the strong calibration performance by demonstrating how the proposed robust training process guarantees to lower the confidence of incorrect predictions. The extensive evaluation shows that the proposed DRE leads to significant calibration improvement without sacrificing the accuracy and burdening inference cost. Furthermore, experiments on OOD and open-set datasets show its effectiveness in terms of generalization and novelty detection capability, respectively.

## Acknowledgement

This research was supported in part by an NSF IIS award IIS-1814450 and an ONR award N00014-18-1-2875. The views and conclusions contained in this paper are those of the authors and should not be interpreted as representing any funding agency.

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

# Supplementary Materials

# Appendix

## Table of Contents

## Organization of Appendix

In this appendix, we first present a table summarizing the major notations used by the main paper in Appendix A. Next, we provide detailed information about the training process and hyperaprameters setting B. We provide the detailed proof of Lemma 1 and Theorem 2 in Section C. After that, we provide additional experimental details and results in Appendix D. Finally, we discuss the broader impacts, limitations, and future work of our DRE technique in Appendix E. The link to the source code can be found in the end of the Appendix.

## A   Summary of Notations

Table 5 below shows the major notations used in the main paper. We further assign each notation into one of four major categories: dataset, DRO formulation, sparse training, and theoretical results.

Table 5: Symbols with Descriptions.

| Symbol Group | Notation | Description |
|---|---|---|
| Dataset | $\mathbf{X}$ | Set of training images |
| | $\mathbf{Y}$ | Set of training class labels |
| | $C$ | Total classes |
| | $\hat{y}$ | Predicted class label |
| | $N$ | Total number of training samples |
| | $D$ | Dimensionality of each data sample |
| DRO | $D_f$ | $f$-divergence |
| | $\eta$ | Parameter controlling size of uncertainty set in DRO framework |
| | $z_n$ | Weight associated with $n^{th}$ data sample |
| Sparse Training | $M$ | Number of sparse sub-networks |
| | $\mathcal{K}$ | Density of the given network |
| | $\Theta$ | Parameter associated with given neural network |
| | $\hat{p}$ | Confidence associated with predicted class |
| | $l(\mathbf{x}_n, \Theta)$ | Loss associated with $n^{th}$ data sample |
| Theoretical Results | $\beta$ | Learning rate of the given network |
| | $P$ | Total number of patches in each data sample |
| | $d$ | Dimensionality of each patch |
| | $\mathbf{v}_{c,l}$ | Major $l^{th}$ feature associated with class c |
| | $L$ | Total number of features in each class class |
| | $D_N^S$ | Collection of single-view data samples |
| | $D_N^M$ | Collection of multi-view data samples |
| | $\cup$ | Collection of features |
| | $H$ | Number of convolution layers |
| | $F_c(\mathbf{x})$ | Logistic output for the $c^{th}$ class for the data sample $\mathbf{x}$ |
| | $\mathcal{P}_{\mathbf{v}_{c,l}}$ | Collection of patches containing feature $\mathbf{v}_{c,l}$ in sample $\mathbf{x}_j$ |
| | $\text{SOFT}_c$ | Softmax output for class $c$ |

## B   Robust Loss Optimization in DRO

In this section, we first provide a detailed description on how we optimize the robust loss function in (1). We then explain how to set the uncertainty set by choosing a proper hyperparameter.

### B.1   Robust Loss Optimization

The optimization problem specified in (1) involves an inequality constraint so directly solving it may incur a higher computational overhead. Therefore, we consider a regularized version of the robust loss to train each base learner by using the following loss:

$$\mathcal{L}^{Robust} = \max_{\mathbf{z} \geq \mathbf{0}, \mathbf{z}^\top \mathbb{1} = 1} \sum_{n=1}^{N} z_n l_n(\Theta) - \lambda D_f \left( \mathbf{z} || \frac{\mathbb{1}}{N} \right) \tag{9}$$

where $l_n(\Theta) = l(\mathbf{x}_n, \Theta)$. Solving the above maximization problem leads to a closed-form solution for $\mathbf{z}^*$ as shown by the following lemma:

**Lemma 3.** *Assuming that $D_f$ is the KL divergence, then solving* (9) *leads to the following solution*

$$\mathcal{L}^{Robust} = \sum_{n=1}^{N} z_n^* l_n(\Theta) \tag{10}$$

*where $z_n^*$ is given by*

$$z_n^* = \frac{\exp\left(\frac{l_n(\Theta)}{\lambda}\right)}{\sum_{j=1}^{N} \exp\left(\frac{l_j(\Theta)}{\lambda}\right)} \tag{11}$$

It can be verified that there is a one-to-one correspondence between $\eta$ in (2) and $\lambda$ in (9). Given their roles in the corresponding equations, a large $\eta$ implies a small $\lambda$ and a small $\eta$ implies a large $\lambda$.

### B.2 Hyperparameter settings

The hyperparameter in the regularization term is chosen based on the difficulty of a dataset. Specifically, for DRE, we always consider the $\lambda \to \infty$ for the first sparse sub-network which is equivalent to Expected Risk Minimization (ERM). For the second and third sub-networks, we choose this hyperparameter based on the difficulty of data samples. It should be noted that we need to set higher $\lambda$ values for more difficult datasets as difficult samples are more common on those datasets. Using this notion, for Cifar10, we choose small $\lambda$ values so that the model can focus on the difficult samples that are few. For this, we choose $\lambda = 10$ for the second sparse sub-network and $\lambda = 500$ for the third sparse sub-network. Considering Cifar100 is more difficult, we would have more difficult samples and therefore higher $\lambda$ value is preferred. For this, we choose $\lambda = 50$ for the second sparse sub-network and $\lambda = 500$ for the third one. In the case of TinyImageNet, we have many difficult samples and therefore we choose relatively large $\lambda$ values. Specifically, we choose $\lambda = 100$ for the second sparse sub-network and $\lambda = 1,000,000$ for the third sparse sub-network.

## C    Proofs of Main Theoretical Results

In this section, we provide detailed proofs of the theoretical results presented in the main paper.

### C.1    Proof of Lemma 1

*Proof.* For $y_n = c$, with respect to data sample $\{\mathbf{x}_n, y_n\}$, the gradient can be evaluated as

$$-\nabla_{\Theta_{c,h}} l(\Theta; \mathbf{x}_n, y_n) = [1 - \text{SOFT}_c(F(\mathbf{x}_n))] \sum_{p \in [P]} \text{ReLU}[\langle \Theta_{c,h}, \mathbf{x}_n^p \rangle] \mathbf{x}_n^p \tag{12}$$

Assume that the given sample has a major feature $\mathbf{v}_{c,l}$, taking dot product with respect to $\mathbf{v}_{c,l}$ on both side of (12) leads

$$\langle -\nabla_{\Theta_{c,h}} l(\Theta; \mathbf{x}_n, y_n), \mathbf{v}_{c,l} \rangle = [1 - \text{SOFT}_c(F(\mathbf{x}_n))] \sum_{p \in [P]} \langle \text{ReLU}[\langle \Theta_{c,h}, \mathbf{x}_n^p \rangle] \mathbf{x}_n^p, \mathbf{v}_{c,l} \rangle \tag{13}$$

Let's further assume that the feature set is orthonormal: $\forall c, c', \forall l \in [L], ||\mathbf{v}_{c,l}||_2 = 1$ and $\mathbf{v}_{c,l} \perp \mathbf{v}_{c',l'}$ when $(c, l) \neq (c', l')$. Using $\mathbf{x}^p = a^p \mathbf{v}_{c,l} + \sum_{\mathbf{v}' \in \cup \setminus \mathbf{v}_c} \alpha^{p, \mathbf{v}'} \mathbf{v}' + \epsilon^p$ given in (4), we have

$$\langle -\nabla_{\Theta_{c,h}} l(\Theta; \mathbf{x}_n, y_n), \mathbf{v}_{c,l} \rangle = [1 - \text{SOFT}_c(F(\mathbf{x}_n))] \left( \sum_{p \in \mathcal{P}_{v,l}(\mathbf{x}_n)} \text{ReLU}[\langle \Theta_{c,h}, \mathbf{x}_n^p \rangle a^p] + \sum_{p \in [P]} \langle \epsilon^p, \mathbf{v}_{c,l} \rangle \right) \tag{14}$$

It should be noted that the term *i.e.,* $\sum_{v' \in \cup \setminus \mathbf{v}_c} \alpha^{p,v'} \langle \mathbf{v}', \mathbf{v}_{c,l} \rangle$ becomes zero due to the orthogonal properties of the feature set. Let us represent the second term by $\kappa$: $\sum_{p \in [P]} \langle \epsilon^p, \mathbf{v}_{c,l} \rangle = \kappa$. Then, we have

$$\langle -\nabla_{\Theta_{c,h}} l(\Theta; \mathbf{x}_n, y_n), \mathbf{v}_{c,l} \rangle = (1 - \text{SOFT}_c(F(\mathbf{x}_n))) \left( \sum_{p \in \mathcal{P}_{v,l}(\mathbf{x}_n)} \text{ReLU}[\langle \Theta_{c,h}, \mathbf{x}_n^p \rangle a^p] + \kappa \right) \tag{15}$$

Furthermore, let us define $V_{c,h,l}(\mathbf{x}_j) = \sum_{p \in \mathcal{P}_{\mathbf{v}_{c,l}}(\mathbf{x}_j)} \text{ReLU}(\langle \Theta_{c,h}, \mathbf{x}_j^p \rangle a^p)$ then above equation further reduces to following

$$\langle -\nabla_{\Theta_{c,h}} l(\Theta; \mathbf{x}_n, y_n), \mathbf{v}_{c,l} \rangle = (1 - \text{SOFT}_c(F(\mathbf{x}_n)))(V_{c,h,l}(\mathbf{x}_n) + \kappa) \tag{16}$$

Recall the above equation is the gradient with respect to the $n^{th}$ data sample. Considering the gradient with respect to all data samples with $y_n = c$, and let us consider the total loss, where the weight $z_n$ of each loss is assigned according to a distribution specified by the uncertainty set $\mathcal{U}$. Then, the total gradient is

$$\langle -\nabla_{\Theta_{c,h}} l(\Theta; \mathbf{X}, \mathbf{Y}), \mathbf{v}_{c,l} \rangle = \max_{\mathbf{z} \in \mathcal{U}} \sum_{n=1}^{N} z_n \left[ \mathbb{1}_{y_j=c} (V_{c,h,l}(\mathbf{x}_n) + \kappa)(1 - \text{SOFT}_c(F(\mathbf{x}_n))) \right] \tag{17}$$

Now using the standard gradient update rule with $\beta$ being the learning rate, we have

$$\langle \Theta_{c,h}^{t+1}, \mathbf{v}_{c,l} \rangle = \langle \Theta_{c,h}^{t}, \mathbf{v}_{c,l} \rangle + \beta \max_{\mathbf{z} \in \mathcal{U}} \sum_{n=1}^{N} z_n \left[ \mathbb{1}_{y_j=c} (V_{c,h,l}(\mathbf{x}_n) + \kappa)(1 - \text{SOFT}_c(F(\mathbf{x}_n))) \right] \tag{18}$$

Let $\mathbf{x}_k \in \mathcal{D}_N^S$ be the most difficult sample having $\mathbf{v}_{c,l}$ as the main feature. Also, consider $\mathbf{x}_n \in \mathcal{D}_N^M$ to be the easy sample with $y_n = c, y_k = c$. Then, we have

$$[1 - \text{SOFT}_c(F(\mathbf{x}_k))] \geq [(1 - \text{SOFT}_c(F(\mathbf{x}_n))], \ \forall n \in [1, N], n \neq k, y_n = c \tag{19}$$

Using above property, we can write the following using (18)

$$\langle \Theta_{c,h}^{t}, \mathbf{v}_{c,l} \rangle + \beta \max_{\mathbf{z} \in \mathcal{U}} \sum_{n=1}^{N} z_n \left[ \mathbb{1}_{y_j=c} (V_{c,h,l}(\mathbf{x}_n) + \kappa)(1 - \text{SOFT}_c(F(\mathbf{x}_n))) \right]$$
$$\leq \langle \Theta_{c,h}^{t}, \mathbf{v}_{c,l} \rangle + \beta N z_k (1 - \text{SOFT}_c(F(\mathbf{x}_k))) \tag{20}$$

On the r.h.s., we have $z_n = \frac{1}{N}$ for ERM, which assigns equal weights to all samples. Under the assumption of $N_{\mathbf{v}_{c,l}} \ll N_{\cup \setminus \mathbf{v}_{c,l}}$, the contribution of the $N_{\mathbf{v}_{c,l}}$ on overall gradient will be negligible. In contrast, for the DRO framework, using (11), we have

$$z_k = \frac{1}{\sum_{j=1, j \neq k}^{N} \exp\left(\frac{l_j(\Theta) - l_k(\Theta)}{\lambda}\right) + 1} \tag{21}$$

Since $l_k(\Theta) > l_j(\Theta), \forall \lambda > 0, \lambda \neq \infty$, we have $z_k > \frac{1}{N}$. Using r.h.s. of (20) and incorporating $z_k = \frac{1}{N}$ for ERM and $z_k > \frac{1}{N}$, we have

$$\{\langle \Theta_{c,h}^{t}, \mathbf{v}_{c,l} \rangle + \beta(1 - \text{SOFT}_c(F(\mathbf{x}_k)))\}_{ERM} \leq \{\langle \Theta_{c,h}^{t}, \mathbf{v}_{c,l} \rangle + \beta(1 - \text{SOFT}_c(F(\mathbf{x}_k)))\}_{Robust} \tag{22}$$

This subsequently leads to the following:

$$\{\langle \Theta_{c,h}^{t}, \mathbf{v}_{c,l} \rangle\}_{Robust} > \{\langle \Theta_{c,h}^{t}, \mathbf{v}_{c,l} \rangle\}_{ERM}; \forall t > 0 \tag{23}$$

which completes the proof of Lemma 1. □

## C.2 Proof of Theorem 2

Let $\mathbf{x} \in \mathcal{D}_S^N$ from class c with $\mathbf{v}_{c,l}$ as the main feature and $\mathbf{v}'$ as the dominant feature learned through the memorization. Also consider $\mathbf{v}'$ to be the main feature characterizing class k. Then for any class $c'$, we can define the following

$$\text{SOFT}_{c'}(\mathbf{x}) = \frac{\exp(F_{c'}(\mathbf{x}))}{\sum_{j \in [C]} \exp(F_j(\mathbf{x}))} \tag{24}$$

In the above equation, $F_{c'}(\mathbf{x})$ can be written as

$$F_{c'}(\mathbf{x}) = \sum_{h \in [H]} \sum_{p \in [P]} \text{ReLU}[\langle \Theta_{c',h}, \mathbf{x}^p \rangle] \tag{25}$$

Substituting $\mathbf{x}^p$ from (4), we have

$$F_{c'}(\mathbf{x}) = \sum_{h\in[H]} \sum_{p\in[P]} \texttt{ReLU}\left[a^p\langle\Theta_{c',h},\mathbf{v}_{c,l}\rangle + \sum_{\mathbf{v}'\in\cup\setminus\mathbf{v}_c} \alpha^{p,\mathbf{v}'}\langle\Theta_{c',h},\mathbf{v}'\rangle + \langle\Theta_{c',h},\epsilon^p\rangle\right] \qquad (26)$$

Substituting $c'$ by $k$, we have

$$F_k(\mathbf{x}) = \sum_{h\in[H]} \sum_{p\in[P]} \texttt{ReLU}\left[a^p\langle\Theta_{k,h},\mathbf{v}_{c,l}\rangle + \sum_{\mathbf{v}'in\cup\setminus\mathbf{v}_c} \alpha^{p,\mathbf{v}'}\langle\Theta_{k,h},\mathbf{v}'\rangle + \langle\Theta_{k,h},\epsilon^p\rangle\right] \qquad (27)$$

In case of ERM, the $\mathbf{v}_{c,l}$ signal is fairly weak during the training process due to $N_{\mathbf{v}_{c,l}} \ll N_{\cup\setminus\mathbf{v}_{c,l}}$. Therefore, the term $\langle\Theta_{k,h},\mathbf{v}_{c,l}\rangle$ is negligible. Also, the last term $\langle\Theta_{k,h},\epsilon^p\rangle$ is also small as this corresponds to the Gaussian noise. For the second term $\exists\mathbf{v}'$ for which $\langle\Theta_{k,h},\mathbf{v}'\rangle$ is very high because of the spurious correlation. In contrast, for the robust loss, using Lemma 1, the model learns a stronger correlation with the true class parameter and therefore $\langle\Theta_{c,h},\mathbf{v}_{c,l}\rangle$ is high. As such, both terms $\langle\Theta_{k,h},\mathbf{v}_{c,l}\rangle$ as well as $\langle\Theta_{k,h},\mathbf{v}'\rangle, \forall v\prime$ becomes low. As a result, we have

$$\{F_k(\mathbf{x})\}_{ERM} > \{F_k(\mathbf{x})\}_{Robust} \qquad (28)$$

Substituting this inequality to (24), we have

$$\{\texttt{SOFT}_k(\mathbf{x})\}_{Robust} < \{\texttt{SOFT}_k(\mathbf{x})\}_{ERM} \qquad (29)$$

This completes the proof of Theorem 2.

# D  Experimental Details and Additional Results

In this section, we first provide a detailed description of datasets used in our experimentation followed by hardware description of our experimentation. We then provide examples of single-view and multi-view data samples. Next, we provide additional experimental results and baselines on Cifar10 and Cifar100 datasets. After that, we provide additional baseline results TinyImageNet. We also compare individual ensemble members' performance followed by our model performance with different calibration techniques commonly used in dense networks. Then, we perform an in-depth ablation study. Parameter size and inference speed are discussed in the subsequent subsection. We also further investigate the diversity of the sparse subnetworks. Finally, we provide detailed qualitative analysis to support our proposed claim.

## D.1  Detailed Dataset Description

For general classification setting, we consider Cifar10, Cifar100 [12], and TinyImageNet [14] datasets. For the out of distribution setting, we consider corrupted version of Cifar10 and Cifar100, which are named as Cifar10-C and Cifar100-C [10], respectively. Finally, for open-set detection, we leverage SVHN [21] as the open-set dataset. The detailed description of each dataset is given below:

- *Cifar10* consists of total 10 classes, each consisting of 5,000 training samples and 1,000 testing (evaluation) samples. Each image is a colored image with size $32 \times 32$.
- *Cifar100* consists of 20 super classes where each super-class consists of 5 classes resulting into total 100 classes. Each class consists of 500 training samples and 100 testing samples. Each image is a colored image with size $32 \times 32$.
- *TinyImageNet* consists of 200 classes with 1,000,000 samples where each class has 500 training images, 50 validation images, and 50 test images. Each image is a colored image with size $64 \times 64$.
- *Cifar10-C* consists of fifteen different types of corruptions applied on the Cifar10 clean testing dataset where each corruption has 5 severity levels, ranging from 1 to 5 with 1 being least severe and 5 being most severe. The corruptions include Gaussian noise, shot noise, impulse noise, defocus blur, forsted glass blur, motion blur, zoom blur, snow, frost, fog, brightness, contrast, elastic, pixelate, and JPEG.
- *Cifar10-C.* consists of fifteen different corruptions applied on the Cifar100 clean testing dataset.
- *SVHN* consists of 10 classes with digit 1 as class 1, digit 9 as class 9 and digit 0 as class 10. These are original, variable-resolution, colored house-number images with character level bounding boxes. We use this dataset as the open-set dataset in our experimentation.

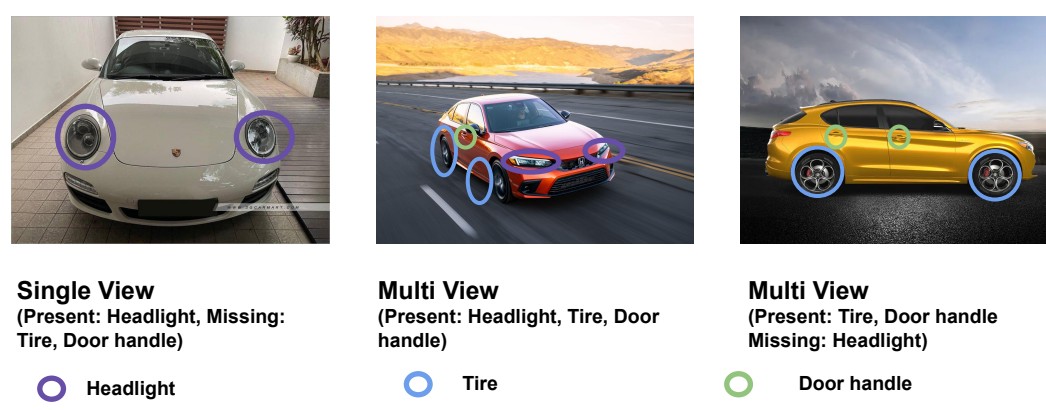

Figure 5: Examples of single-view and multi-view samples.

## D.2 Hardware Details for Experimentation

All experimentations are conducted using NVIDIA RTX A6000 GPU with 48GB memory requiring 300 Watt power. For GPU, CUDA Version: 11.6, Driver Version: 510.108.03, and NVIDIA-SMI: 510.108.03 is used. In terms of CPU, our experimentation uses an Intel(R) Xeon(R) Gold 6326 CPU @ 2.90GHz with a 64-bit system and an x86_64 architecture.

## D.3 Single-view and Multi-view Examples

Figure 5 show the three example images, where the first image is a representative single-view data sample whereas the last two are multi-view samples. In this example, we consider three major features for cars: *i.e.,* `Tire`, `Headlight`, and `Door handle`. As only headlight feature is present in the first image, it belongs to the single-view category. For the second and third images, multiple features are presented and therefore we regard those images as multi-view data samples.

Table 6: Accuracy and ECE performance with $15\%$ density for Cifar10 and Cifar100 dataset.

| Training Type | Approach | Cifar10 | | | | Cifar100 | | | |
| | | ResNet50 | | ResNet101 | | ResNet101 | | ResNet152 | |
| | | $ACC$ | $ECE$ | $ACC$ | $ECE$ | $ACC$ | $ECE$ | $ACC$ | $ECE$ |
|---|---|---|---|---|---|---|---|---|---|
| | Dense† | 94.82 | 5.87 | 95.12 | 5.99 | 76.40 | 16.89 | 77.97 | 16.73 |
| Dense Training | L1 Pruning | 93.88 | 5.69 | 94.23 | 5.88 | 75.53 | 15.52 | 75.83 | 15.78 |
| | LTH | 92.97 | 4.03 | 93.15 | 5.69 | 74.36 | 15.13 | 74.77 | 15.22 |
| | DLTH | 95.15 | 6.21 | 95.65 | 6.96 | 77.98 | 16.24 | 78.23 | 16.54 |
| | Mixup | 93.22 | 4.02 | 93.38 | 5.68 | 74.48 | 15.10 | 74.68 | 15.16 |
| Sparse Training | CigL | 92.25 | 4.67 | 93.34 | 4.59 | 77.88 | 10.16 | 77.27 | 10.62 |
| | DST Ensemble | 89.57 | 2.10 | 88.64 | 1.34 | 64.57 | 9.76 | 64.75 | 9.27 |
| | Sup-ticket | 94.65 | 3.20 | 94.95 | 3.09 | 78.68 | 10.16 | 78.95 | 10.32 |
| Mask Training | AdaBoost | 94.07 | 5.65 | 94.76 | 5.14 | 75.98 | 23.55 | 76.28 | 24.27 |
| | EP | 94.41 | 3.90 | 94.42 | 4.07 | 75.66 | 14.79 | 76.05 | 14.79 |
| | SNE | 94.85 | 3.05 | 94.96 | 3.18 | 76.82 | 11.12 | 77.23 | 11.63 |
| | **DRE** | 94.87 | **1.71** | 94.74 | **1.34** | 75.86 | **4.90** | 76.46 | **5.81** |

## D.4 Additional Result on Cifar10 and Cifar100

Table 6 shows the experimental result on Cifar10 and Cifar100 datasets with a $15\%$ density. As shown, the proposed technique has a far superior performance in terms of the ECE score compared to the competitive baselines. This is consistent with the results with a 9% density as presented in the main paper, which further justifies the effectiveness of our proposed technique.

### D.5 Additional Baseline Results on Sparse Training Methods

Apart from the baselines included in the main paper, there are other sparse training methods without the need for iterative pruning/growing [15, 19, 33, 36]. However, as all these methods primarily focus on pushing the accuracy up to the original dense networks, they still suffer from a severely overfitting behavior, leading to a poor calibration performance as shown in Table 7.

Table 7: ECE and Accuracy for other sparse baselines.

| Approach | Cifar10 | | | | Cifar100 | | | |
| | 9% | | 15% | | 9% | | 15% | |
| | $\mathcal{ACC}$ | $\mathcal{ECE}$ | $\mathcal{ACC}$ | $\mathcal{ECE}$ | $\mathcal{ACC}$ | $\mathcal{ECE}$ | $\mathcal{ACC}$ | $\mathcal{ECE}$ |
|---|---|---|---|---|---|---|---|---|
| SNIP [15] | 93.45 | 4.10 | 94.12 | 3.86 | 52.99 | 10.97 | 54.40 | 10.56 |
| Dynamic Sparse [19] | 91.11 | 5.73 | 92.09 | 5.33 | 70.06 | 15.56 | 71.23 | 14.29 |
| GraSP [33] | 92.73 | 4.78 | 93.16 | 4.54 | 72.60 | 17.07 | 73.07 | 15.95 |
| MEST [36] | 92.94 | 4.67 | 93.50 | 4.40 | 72.19 | 16.47 | 73.49 | 15.81 |
| EP | 94.20 | 3.97 | 94.41 | 3.90 | 75.07 | 14.62 | 75.66 | 14.79 |
| ***DRE*** | 94.60 | **0.7** | 94.87 | **1.71** | 74.68 | **1.20** | 75.86 | **4.90** |

### D.6 Additional Baseline Results on TinyImageNet

As mentioned in the main paper, the computational issue (*i.e.,* memory overflow) makes it impossible to run sparse learning techniques *i.e.,* CigL [16], DST Ensemble [18], and Sup-ticket [35] on the ResNet101 and WideResNet101 architectures to make a fair comparison. Therefore, in this section, we pick a lower capacity model (ResNet50) and compare the performance. Even for the ResNet50 architecture, CigL still runs into the memory overflow issue with a batch

Table 8: Additional baseline results on Tiny-ImageNet using ResNet50 with $\mathcal{K} = 15\%$.

| Training Type | Approach | $\mathcal{ACC}$ | $\mathcal{ECE}$ |
|---|---|---|---|
| Sparse Training | *DST Ensemble* | 72.00 | 2.94 |
| | *Sup-ticket* | 68.68 | 10.96 |
| Mask Training | *DRE* | 71.57 | 1.51 |

size of 128. Furthermore, lowering the batch size (*e.g.,* 16) makes the training process extremely slow even using a $48Gb$ GPU, where each training epoch takes more than half an hour, making model training extremely difficult. Therefore, we did not report the performance of CigL. It should be noted that CigL can be trained on Cifar10 and Cifar100 because of lower dimension of the input images and we have already reported its performance in the main paper. Table 8 shows the performance of DRE along with those from DST Ensemble and Sup-ticket on ResNet50. It is clear that DRE achieves better performance compared to these baselines.

### D.7 Performance from Ensemble Members

We investigate how performance varies in different sparse sub-networks. We use Cifar100 as an example and Table 9 report the individual sub-network performance on both accuracy and ECE. While each sparse sub-network is a relatively weaker learner (which is expected), they contribute to the final ensemble model in a complementary way, leading to a better ECE score as well as accuracy.

Table 9: Different sub-networks performance on Cifar100 dataset.

| Subnetworks | ResNet101 | | ResNet152 | |
| | $\mathcal{ACC}$ | $\mathcal{ECE}$ | $\mathcal{ACC}$ | $\mathcal{ECE}$ |
|---|---|---|---|---|
| *Subetwork 1 (3%)* | 68.22 | 14.35 | 69.65 | 13.31 |
| *Subetwork 2 (3%)* | 69.03 | 1.39 | 70.00 | 3.39 |
| *Subetwork 3 (3%)* | 72.86 | 11.96 | 70.24 | 14.78 |
| *DRE* | 74.68 | **1.20** | 74.37 | 2.09 |

### D.8 Comparison with Common Calibration Techniques

In this section, we investigate whether existing calibration techniques designed for training dense networks can be leveraged to further improve the calibration performance of sparse networks. However, most of these techniques (*e.g.,* temperature scaling and mix-n-match) are post hoc techniques, which require a separate validation set to fine-tune the parameters. This means we need to further divide the training data into training and validation sets, which may negatively impact the generalization capability of the trained model (due to less training data). To make a comparison, we pick Temperature Scaling (TS) [9], Label Smoothing (LS) [31], and a few other techniques proposed in

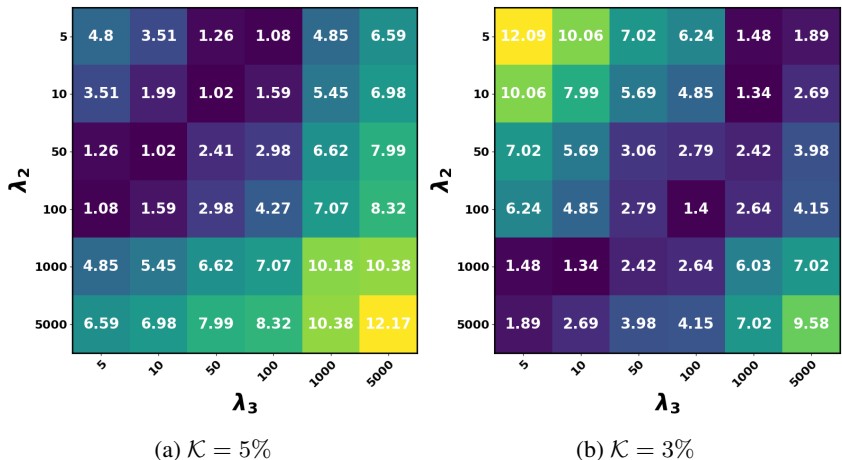

(a) $\mathcal{K} = 5\%$                    (b) $\mathcal{K} = 3\%$

Figure 6: (a-b) Impact of $\lambda$ on ECE using ResNet101 architecture on Cifar100 dataset.

[37], including Ensemble Temperature Scaling (ETS) and Isotonoic Regression One vs All combined with Temperature Scaling (IROvA-TS). We apply these calibration techniques on the top of the EP algorithm. Specifically, as LS does not require a separate validation set, we train it on the full training dataset using the LS loss (with $\epsilon = 0.1$). Other calibration techniques require a separate validation set and therefore we divide training data into training and validation with a 80:20 ratio. EP (No Validation) uses the full training dataset whereas EP (Validation) is trained using $80\%$ of the training data. Once the model is trained with $80\%$ of training data using EP, we further calibrate it using the aforementioned calibration techniques. Table 10 shows the results. There are two key observations: (i) the classification accuracy decreases for all calibration techniques at the expense of improving calibration performance as they require a separate validation set, and (ii) DRE achieves the best ECE in all cases, which further justifies its strong calibration performance.

Table 10: Different calibration techniques on the top of EP Algorithm with $\mathcal{K} = 9\%$.

| | Cifar10 | | | | Cifar100 | | | |
|---|---|---|---|---|---|---|---|---|
| Approach | ResNet50 | | ResNet101 | | ResNet101 | | ResNet152 | |
| | $\mathcal{ACC}$ | $\mathcal{ECE}$ | $\mathcal{ACC}$ | $\mathcal{ECE}$ | $\mathcal{ACC}$ | $\mathcal{ECE}$ | $\mathcal{ACC}$ | $\mathcal{ECE}$ |
| TS | 93.42 | 0.96 | 93.42 | 1.37 | 73.06 | 1.72 | 73.40 | 2.45 |
| ETS | 93.42 | 0.97 | 93.42 | 1.37 | 73.06 | 1.76 | 73.40 | 2.40 |
| IROvA-TS | 89.90 | 1.45 | 88.69 | 0.89 | 60.87 | 1.56 | 60.77 | 2.86 |
| LS | 94.06 | 7.56 | 94.21 | 7.41 | 75.96 | 9.36 | 76.40 | 7.71 |
| EP (No Validation) | 94.20 | 3.97 | 94.35 | 4.03 | 75.05 | 14.62 | 75.68 | 14.41 |
| EP (Validation) | 93.42 | 4.46 | 93.42 | 4.83 | 73.06 | 15.56 | 73.40 | 15.88 |
| *DRE* | 94.60 | **0.7** | 94.28 | **0.7** | 74.68 | **1.20** | 74.37 | **2.09** |

## D.9 Ablation Study

In this section, we first show the impact of $\lambda$ values on the prediction and calibration performance. We then compare the performance of proposed DRE with respect to dense ensemble techniques. Finally, we show how proposed DRE avoids learning from noisy features by considering the dataset containing explicit spurious features.

**Impact of the uncertainty set size.** For simplicity, we always keep one sparse sub-network in our framework to be with $\lambda_1 \to \infty$. The ECE performance with respect to different sets of $\lambda$ value for the remaining sub-networks is shown using the heatmap given in Figure 6 (a-b). As can be seen, it is important to choose $\lambda_2$ and $\lambda_3$ with very distinct values to achieve a low calibration error.

**Sparse Ensemble vs Dense Ensemble.** To more clearly demonstrate the advantage of using the proposed DRE compared to the dense ensemble, we have conducted additional experimentation and present the results in Table 11. The Dense Ensemble (w/o DRO) refers to the one, where we ensemble multiple dense networks and each one is trained using the standard ERM loss. The Dense Ensemble (w/ DRO) is the one, where we train multiple dense networks but using the DRO loss given by Eq. (1). As can be seen, the proposed sparse ensemble (*i.e.,* DRE) clearly outperforms the dense ensemble to a large extent. It should be noted that a dense ensemble (with DRO) only achieves a slightly better ECE score as compared with dense ensemble (w/o DRO). This is because, it is more difficult to further diversify different a dense networks with the exactly same architectures (*i.e.,* nodes and connections). In contrast, using sparse training, we can naturally pick very distinct sparse subnetworks from the original dense network to increase the diversity, where each subnetwork is already better calibrated because of the reason explained above. Additionally, thanks to the distributionally robust ensemble, we can further diversify the learned subnetworks leading to much better calibration performance. This result also helps to further justify our key motivation for developing sparse ensembles by showing how dense networks are poorly calibrated, resulting from the memorization effect introduced by the over-parameterized architecture.

Table 11: ECE and Accuracy comparison between dense ensemble and DRE.

| Approach | Cifar10 | | | | Cifar100 | | | |
| --- | --- | --- | --- | --- | --- | --- | --- | --- |
| | ResNet50 | | ResNet101 | | ResNet101 | | ResNet152 | |
| | $\mathcal{ACC}$ | $\mathcal{ECE}$ | $\mathcal{ACC}$ | $\mathcal{ECE}$ | $\mathcal{ACC}$ | $\mathcal{ECE}$ | $\mathcal{ACC}$ | $\mathcal{ECE}$ |
| Dense Ensemble (w/o DRO) | 94.99 | 3.95 | 95.29 | 3.69 | 78.38 | 9.11 | 78.40 | 9.01 |
| Dense Ensemble (w/ DRO) | 94.35 | 3.90 | 94.30 | 2.97 | 77.69 | 8.23 | 77.79 | 7.57 |
| *DRE* | 94.87 | **1.71** | 94.74 | **1.34** | 75.86 | 4.90 | 76.46 | **5.81** |

Table 12: Training, validation, and testing data distribution in the waterbird dataset.

| Background | Training | | Validation | | Testing | |
| --- | --- | --- | --- | --- | --- | --- |
| | Waterbird | Landbird | Waterbird | Landbird | Waterbird | Landbird |
| Waterbackground | 1057 | 184 | 133 | 466 | 642 | 2225 |
| Landbackground | 56 | 3498 | 133 | 467 | 642 | 2255 |

**Performance with respect to explicit spurious features.** To demonstrate how our proposed approach improves the calibration by avoiding the noisy (spurious) features, we conduct additional experiments on the waterbird dataset [26], which contains explicit spurious correlations. Specifically, in this dataset, there are two classes: (a) waterbird and (b) landbird. Most of the waterbirds images are taken in the water background whereas landbirds images are taken in the land background. Hence, the model will have a tendency to make an association between the background and the type of bird instead of focusing on the true underlying features of the birds (*e.g.,* color of the feature, *etc.*). Table 12 summarizes the the data distribution. There are limited data samples without the spurious correlation in the training set and therefore the model is likely to predict based on the background instead using the true features. Compared to the training set, the validation and testing sets are less skewed and therefore evaluation on testing set will no longer be favored by only focusing on the spurious correlation.

Table 13 shows the performance from the sparse network ensemble SNE and DRE with a 15% total sparsity. In the table, Original indicates the original testing set provided in Table 12, whereas Spurious Only is the one where only data samples with spurious correlations (*i.e.,* waterbird on a water background and landbird on a land background) are considered and Non-spurious only indicates the samples without spurious

Table 13: Performance on waterbird dataset.

| Approach | Original | | Spurious Only | | Non-spurious Only | |
| --- | --- | --- | --- | --- | --- | --- |
| | $\mathcal{ACC}$ | $\mathcal{ECE}$ | $\mathcal{ACC}$ | $\mathcal{ECE}$ | $\mathcal{ACC}$ | $\mathcal{ECE}$ |
| *SNE* | 77.64 | 13.78 | 85.95 | 9.50 | 64.17 | 24.89 |
| *DRE* | 77.83 | 8.32 | 84.76 | 8.67 | 66.30 | 19.96 |

correlations (*i.e.,* waterbird on a land background and a landbird on a water background). There are two key observations. First, SNE performs similarly with the DRE in case of samples holding spurious correlations (*i.e.,* spurious only) where the overconfident predictions are usually preferred as they are most likely to be correct benefiting from the spurious correlation. Second, in the case

of non-spurious only, DRE achieves better performance both in terms of accuracy and ECE, which justifies that our model indeed learns from important features instead of spurious correlations. In the original test set, because of the large number of samples holding spurious correlations, we do not see a clear advantage in terms of accuracy. However, DRE still achieves a clearly better calibration performance compared to SNE.

We have further visualized the the heatmap of convolution 4 layer using the Grad-Cam technique. As shown in the Figure 7 (a), the sparse sub-network in the SNE focuses on the water background instead of focusing on the actual landbird object. This is because, during training process, the sparse sub-network in SNE is likely to learn to associate the spurious background feature with the true label. Specifically, the model learns to predict landbird whenever there is a land background and waterbird whenever there is a water background. In contrast, using the DRE technique, as demonstrated through the heatmap, the sparse sub-network focuses on the actual object instead of the background. It is worth mentioning that, because of the overfitting phenomenon and lack of a systematic way for diversification, each sparse sub-network in the SNE behaves in a similar way by focusing on the spurious feature instead of the actual object. In contrast, in the case of DRE, each sparse sub-network is controlled by the $\eta$ parameter in Eq. (2). Specifically, we use a low $\eta$ value (*i.e.,* $\eta \rightarrow 0$) for one of the sparse sub-networks, which will be similar to that of the sparse sub-network obtained using the SNE. However, for the higher $\eta$ value, it will focus on learning from more difficult samples, including those not holding the spurious correlations. As such, the sparse sub-network is forced to learn from the actual object instead of through the background. Therefore, the model focuses mostly on actual objects as demonstrated in Figure 7 (b). When these diverse sparse sub-networks are combined in the DRE, it achieves a better calibration without being confidently wrong like in the SNE.

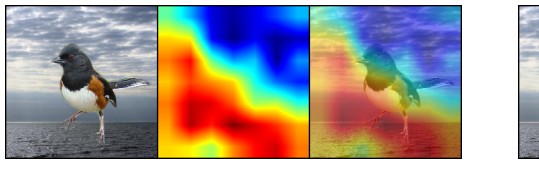 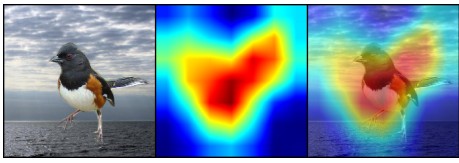

(a) SNE sparse sub-network            (b) DRE sparse sub-network

Figure 7: Visualization of the layer 4 convolution using the Grad-Cam technique [29] on a landbird data sample: (a) SNE focuses on the water background, (b) DRE focuses on the actual object.

### D.10    Parameter Size and Inference Speed

We compare parameter size and inference speed of different types of sparse networks. Table 14 shows the FLOPS along with number of parameters associated with each technique. As can be seen, the proposed DRE has a comparable parameter size as that of the sparse network ensemble. In terms of computational times, our

Table 14: Parameter size and inference speed.

| Approach | ResNet50 | | ResNet101 | |
|---|---|---|---|---|
| | Params | Flops ($\times 10^9$) | Params | Flops ($\times 10^9$) |
| *Dense*[†] | 23.6M | 4.14 | 42.5M | 7.88 |
| *SNE* | 3.5M | 1.31 | 6.3M | 2.53 |
| *DRE* | 3.5M | 1.31 | 6.3M | 2.53 |

approach is comparable to the sparse network ensemble. Compared to a dense network, our technique has a much smaller parameter size with less FLOPS.

### D.11    Diversity on Sparse Sub-networks

To justify our claim that our technique ensures the diverse sparse sub-networks, we adapt the disagreement metric ($d_{dist}$) from [18]. This metric measures the disagreement among sub-networks in terms of class label prediction. Table 15 below shows the results for Cifar10 and Cifar100 datasets. As shown, compared to Sparse Network Ensemble, DRE achieves higher disagreement which implies that the sparse sub-networks are more diverse.

### D.12    Qualitative Analysis

As demonstrated in the qualitative analysis section of the main paper, our approach is able to generate the less confident prediction in the case of incorrectly samples and thereby making model better calibrated. Figure 8 (a)-(d) show the confidence values of the wrongly classified samples from

Table 15: Accuracy, ECE, and prediction disagreement performance with a $\mathcal{K} = 15\%$ density.

| Approach | Cifar10 | | | | | | Cifar100 | | | | | |
|---|---|---|---|---|---|---|---|---|---|---|---|---|
| | ResNet50 | | | ResNet101 | | | ResNet101 | | | ResNet152 | | |
| | $\mathcal{ACC}$ | $\mathcal{ECE}$ | $d_{dist}$ | $\mathcal{ACC}$ | $\mathcal{ECE}$ | $d_{dist}$ | $\mathcal{ACC}$ | $\mathcal{ECE}$ | $d_{dist}$ | $\mathcal{ACC}$ | $\mathcal{ECE}$ | $d_{dist}$ |
| SNE | 94.85 | 3.05 | 0.048 | 94.96 | 3.18 | 0.049 | 76.82 | 11.12 | 0.20 | 77.23 | 11.63 | 0.20 |
| *DRE (Ours)* | 94.87 | **1.71** | 0.088 | 94.74 | **1.34** | 0.069 | 75.86 | **4.90** | 0.24 | 76.46 | **5.81** | 0.24 |

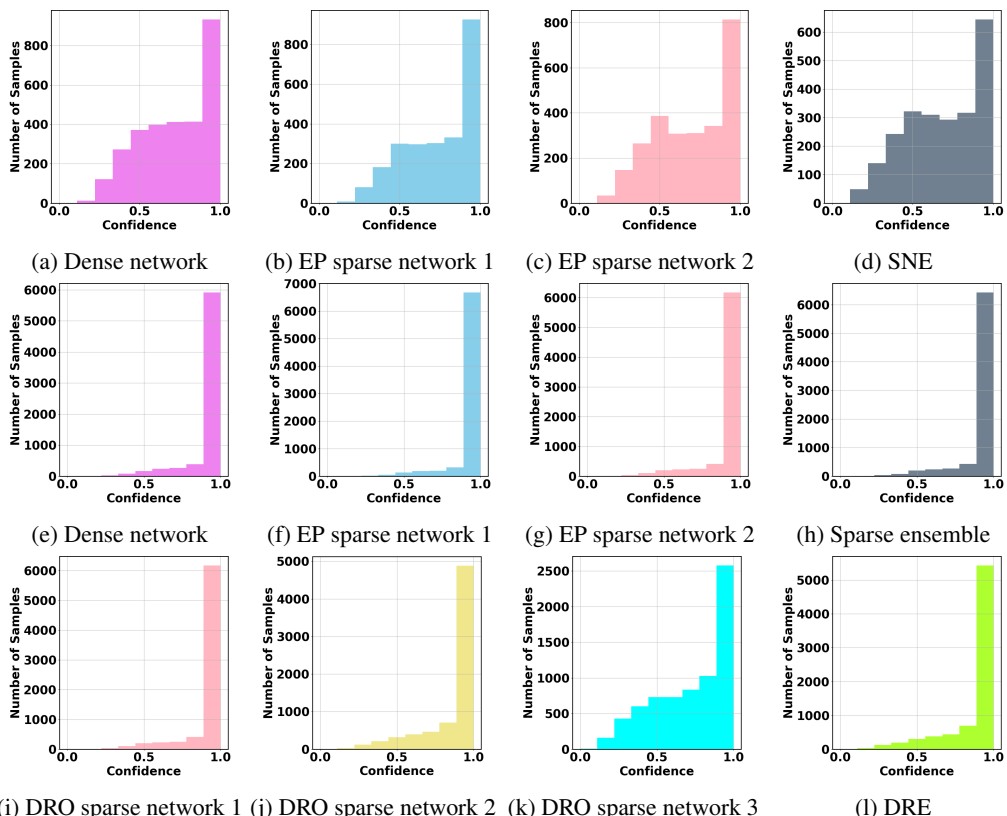

(a) Dense network   (b) EP sparse network 1   (c) EP sparse network 2   (d) SNE

(e) Dense network   (f) EP sparse network 1   (g) EP sparse network 2   (h) Sparse ensemble

(i) DRO sparse network 1 (j) DRO sparse network 2 (k) DRO sparse network 3   (l) DRE

Figure 8: Confidence scores of correctly classified samples in baseline models (a)-(d); confidence scores of correctly classified samples in CIFAR100 with ResNet101: (e)-(l)

the baseline models, which are concentrated on the higher end as compared with DRE. We further show that DRE behavior in the correctly classified samples in Figure 8 (e)-(l). As can be seen, the confidence score of correctly classified data samples from the CIFAR100 dataset with different techniques. As shown, our DRE technique remains confident on the correct data samples while being not confident on the incorrect data samples. This result shows our approach is well calibrated and trustworthy compared with the competitive baselines. In summary, our proposed technique remains uncertain for incorrect samples while being confident on the correct samples resulting in a much improved calibration.

# E   Broader Impact, Limitations, and Future Work

In this section, we first describe the potential broader impacts of our work. We then discuss the limitations and identify some possible future directions.

### E.1 Broader Impact

Sparse network training provides a highly promising way to significantly reduce the computational cost for training large-scale deep neural networks without sacrificing their predictive power. Besides energy savings, it also opens the gate for deploying deep neural networks to lightweight computing or edge devices that can further broaden the applications of AI in more diverse and resource constrained settings. The proposed robust ensemble framework provides a general solution to achieve calibrated training of deep learning models. As a result, the trained model is expected to provide more reliable uncertainty predictions, which could be an important step towards using AI in safety-critical domains.

### E.2 Limitations and Future Works

As an ensemble model, DRE involves multiple base learners (*i.e.,* sparse sub-networks). Consequently, it may lead to more computational overhead. This could create issues for real-time application as during the inference time, the input needs to be passed through all base learners to get the final output, which can slow down the prediction speed. A straightforward way to speed up the inference process is to execute all the base learners in parallel, which still incurs additional computational overhead. One interesting future direction is to investigate knowledge distillation and train a single sparse network from the ensemble model. Theoretical evidence [1] shows that knowledge distillation has the potential to largely maintain the ensemble performance while providing a promising way to train a single sparse network with an even higher sparsity level and improved inference speed.

## F   Source Code

For the source code of this paper, please click here.

