# Supplementary Material

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

Table 9: ACC and ECE with different: (a) backbones and (b) number of subnetworks.

| Approach | WideResNet28-10 | | ViT | |
|---|---|---|---|---|
| | $\mathcal{ACC}$ | $\mathcal{ECE}$ | $\mathcal{ACC}$ | $\mathcal{ECE}$ |
| *EP* | 94.12 | 4.53 | 86.16 | 10.01 |
| *DRE* | 93.98 | **1.93** | 85.53 | **4.18** |

| Approach | ResNet101 | | ResNet152 | |
|---|---|---|---|---|
| | $\mathcal{ACC}$ | $\mathcal{ECE}$ | $\mathcal{ACC}$ | $\mathcal{ECE}$ |
| *DRE (M = 3)* | 94.87 | 1.71 | 94.74 | 1.34 |
| *DRE (M = 5)* | 94.79 | 0.84 | 94.69 | 0.62 |

(a) Different backbones on Cifar10 Dataset.      (b) Different $M$ values on Cifar10 with $\mathcal{K} = 15\%$.

## D.8 Ablation Study

In this section, we first show the impact of $\lambda$ values on the prediction and calibration performance. We then investigate how the size of the ensemble affects it calibration performance. Finally, we show the effectiveness of the proposed technique as we vary the backbones. In addition to the backbones used in the main paper, we will further evaluate two other commonly used backbones, including WideResNet28 and Vision Transformer (ViT) [5] as backbones.

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

## F   Source Code

For the source code of this paper, please click here.

## G   References

[1]. Dosovitskiy et al. An Image is Worth 16x16 Words: Transformers for Image Recognition at Scale. ICLR2021.