# OpenReview forum: "Distributionally Robust Ensemble of Lottery Tickets Towards Calibrated  Sparse Network Training"
_NeurIPS.cc/2023/Conference — NeurIPS 2023 poster_

### Official Review · Reviewer_fvyR · 2023-07-03

**Soundness:** 3 good
**Presentation:** 2 fair
**Contribution:** 2 fair
**Rating:** 5
**Confidence:** 4

**Summary:**

- The authors proposed a novel Distributionally Robust Optimization (DRO) framework to achieve an ensemble of lottery tickets toward calibrated network sparsification.
- The proposed DRO ensemble aimed to learn multiple diverse and complementary sparse sub-networks with the guidance of uncertainty sets, which encourage winning tickets to capture different data distributions from easy to hard gradually.
- The authors theoretically justified the strong calibration performance by showing how the proposed DRO guarantees to lower the confidence of incorrect predictions.
- Extensive experimental results on several benchmarks demonstrated that the proposed DRO leads to a clear calibration improvement without sacrificing accuracy and burdening inference costs.

**Strengths:**

(+) The authors proposed a novel sparse ensemble framework (DRO) that combines competitive sparse sub-networks to achieve better calibration performance with the scheduling of the learning of complementary ensemble sub-networks (tickets).

(+) The proposed robust training process guaranteed to lower the confidence of incorrect predictions and strong calibration performances.

(+) Extensive empirical results demonstrated the proposed lottery ticket ensemble's effectiveness in competitive classification and open-set detection.

**Weaknesses:**

(-)The proposed DRO conducted experiments using (M==3) subnetworks and showed each sub-networks performance and confidence scores (Appendix). However, there is no ablation study on these subnetworks regarding V or V’ feature representations. These representations could help understand the subnetwork’s ensemble better.

(-) Line 258: (Regarding feature representations), is there any evidence or observation that the ERM model learns to memorize some noisy feature v’ introduced through specific spurious correlations?

**Questions:**

(-) line 155 (typos): to follow a multivariate Gaussian distribution with the red dot representing its mean? → gray dot?
(-) line 164: I could not understand the following sentence: “Starting from the second sub-network, the training distribution is changed according to the losses.”

(-) The subnetwork’s feature-level analysis could help understand this work.

**Limitations:**

Please, see above the weaknesses and questions.

---

> ### Author Rebuttal · Authors · 2023-08-09
>
> We would like to thank the reviewer for the valuable comments/suggestions. We summarize our responses as follows.
>
> **Q1: Ablation study regarding V or V’ feature representations.**
>
> Thank you for this great suggestion! Following the reviewer's idea, we conduct additional experiments on the Waterbird dataset [1], which contains explicit spurious correlations. Specifically, in this dataset, there are two classes: (a) waterbird and (b) landbird. Most of the waterbirds images are taken in the water background whereas landbirds images are taken in the land background. Hence, the model will  have a tendency to make an association between the background and the type of bird instead of focusing on the true underlying features of the birds (e.g., color of the feature, etc).
> The Table 3 (in the **attached pdf of the general rebuttal**) summarizes the the data distribution. There are limited data samples without the spurious correlation in the training set and therefore the model is likely to predict based on the background instead using the true features. Compared to the training set, validation and testing sets  are less skewed and therefore evaluation on testing set will no longer be favored by only focusing on the spurious correlation.
>
>
>
>
> The Table 4 (b) (in the **attached pdf of the general rebuttal**)  shows the performance from  the sparse network ensemble SNE and DRE with a 15\% total sparsity. In the table, Original indicates the original testing set provided in  Table 3 of the attached pdf file  whereas, spurious is the one where only data samples with spurious correlations (i.e., waterbird on a water background and land bird on a land background) are considered and non-spurious only indicates the samples without spurious correlation (i.e., waterbird on a land background and a landbird on water background). There are two key observations. First, SNE performs similarly with the DRE in case of samples holding spurious correlations (i.e., spurious only) where the overconfident predictions are usually preferred as they are most likely to be correct benefiting from the spurious correlation. Second, in the case of non-spurious only, DRE achieves better performance both in terms of accuracy and ECE, which justifies that our model indeed learns from important features instead of spurious correlations. In the original test set, because of the large number of samples holding spurious correlation, we do not see a clear advantage in terms of accuracy. However, DRE still achieves a clearly better calibration performance compared to SNE.
>
>
>
>
>
>
>
> **Q2: Evidence or observation that the ERM model learns to memorize some noisy feature v’ introduced through specific spurious correlations**
>
> Thanks you for this insightful comments. In fact, we have provided the evidence that the ERM model learns to memorize noisy features introduced through the spurious correlations in Figure 6 of Appendix D.11. We show the number of incorrectly classified samples with respect to confidence score using different techniques. Plots a-d use the ERM model, where a majority of samples are concentrated in the high confidence region despite being incorrect. This is because, the model learns to pick the noisy features (i.e., spurious correlations which cause overfitting) in addition to the important learning signals during the training process. As such, during testing, whenever such noise features occur, the model produces overconfident predictions while ignoring the true signals. In contrast, using DRE (plot h) and DRO (plots f and g), the model becomes less confident in those wrong cases, and therefore a majority of the wrong samples are concentrated in the low-confidence region. This is because the DRO technique forces the model to learn from the important signals instead of spurious correlations. As such, the model does not learn to produce many overconfident predictions as by ERM.
>
> **Q3: Clarity on sentence in line 164**
>
>
> In the case of AdaBoost, we start by training the first sub-network. Then, based on its performance, we assign the importance of each data sample (based on their loss) and train the second sub-network. This means, a higher loss resulting from the first sub-network results in a larger weight to the given sample for the second sub-network, where weight indicates the probability of a given data point being sampled during training. In other words, difficult samples will appear more frequently in the second sub-network compared to the easier one. To train the third sub-network, we again assign higher weights to samples with higher losses from the second sub-network and perform training based on new data distribution based on new weights. We repeat this process. We will make this clear in the revised paper.
>
> **Q4: Typo**
>
> Thanks for carefully checking our paper and identifying the typo. We will fix the issue and improve the presentation of the revised paper.
>
> **Q5: Subnetwork's feature-level analysis**
>
> This is a great suggestion! We have visualized the features of a (non-DRO) subnetwork chooses to focus and made a comparison with features focused by a subnetwork trained through the DRO framework. Please refer to Figure 1 in **the pdf file attached with the general rebuttal**.
>
>
> **References**
>
> - [1] Sagawa et al. "Distributionally Robust Neural Networks for Group Shifts: On the Importance of Regularization for Worst-Case Generalization". ICLR2020.

---

> > ### Comment · Reviewer_fvyR · 2023-08-16
> >
> > Thank the author for the detailed rebuttals.
> >
> > The authors clarified the subnetwork's feature-level analysis through additional ablation studies. However, I wanted to observe the subnetwork-wise representation instead of the subnetwork's representation because this observation would support this work's motivation and effectiveness of ensemble subnetworks.
> >
> > So, I would decrease my score to borderline accept.

---

> > > ### Author Response · Authors · 2023-08-17
> > >
> > > We would like to thank the reviewer for going through our rebuttal. We would be grateful if the reviewer can further clarify the expected result for the 'subnetwork-wise representation'.
> > >
> > > In Figure 1 of attached pdf file we have visualized the the heatmap of convolution 4 layer using the Grad-Cam technique. As shown in the figure, the sparse subntwork in the SNE focuses on the water background instead of focusing on the actual landbird object. This is because, during training process, the sparse subnetwork in SNE is likely to learn to associate the spurious  background feature with the true label. Specifically, the model learns to predict landbird whenever there is a land background and waterbird whenever there is a water background. In contrast, using the DRE technique, as demonstrated through the heatmap, the sparse subnetwork focuses on the actual object instead of the background. It is worth mentioning that, because of the overfitting phenomenon and lack of a systematic way for diversification, each sparse subnetwork in SNE behaves in a similar way by focusing on the spurious feature instead of the actual object. In contrast, in the case of DRE, each sparse subnetwork is controlled by the $\eta$ parameter in Eq. 2 (main paper). Specifically, we use a low  $\eta$ value (i.e.,  $\eta\rightarrow 0$) for one of the sparse subnetworks, which will be similar to that of the sparse subnetwork in the SNE. However, for the higher $\eta$ value, it will focus on learning from more difficult samples, including those not holding the spurious correlation. As such, the sparse subnetwork is forced to learn from the actual object instead of through the background. Therefore, the model focuses mostly on actual objects as demonstrated in Figure 1 (b) of the attached pdf file. When we combine these diverse sparse subnetworks in DRE, we will have a better calibration without being confidently wrong like in SNE. In the revised paper, we will add all subnetworks' heatmaps.
> > >
> > > We hope this can address the reviewer's question about 'subnetwork-wise representation' and we are happy to provide any additional details if needed.

---

### Official Review · Reviewer_RrYi · 2023-07-06

**Soundness:** 3 good
**Presentation:** 2 fair
**Contribution:** 2 fair
**Rating:** 3
**Confidence:** 5

**Summary:**

In this paper, the author proposes a Distributionally Robust Optimization (DRO) framework, which utilizes the ensemble of multiple sparse sub-networks to improve the network calibration. The author argues that the previous ensemble method, i.e., AdaBoost, will make the sub-network severely underfit the training data, leading to a rather poor generalization capability. To solve this problem, the author proposes Distributionally robust ensemble (DRE) method to obtain complementary sparse sub-networks.

**Strengths:**

* The proposed method is evaluated on different datasets and networks.

* The proposed method achieves better accuracy than the baseline works compared.

**Weaknesses:**

* The author argues that the current sparse training works (LTH and EP) have two limitations, which are: (1) the requirement of pretraining a dense network and (2) the learning objective remains as improving the accuracy up to the original dense networks.
However, the author seems completely ignore the very popular (static/dynamic) sparse training methods, which directly train a sparse network from scratch and does not require extra training epochs for iterative pruning and growing, e.g., [1] [2] [3] [4] [5]. It seems that the author argued limitations are not in those sparse training methods.

[1] SNIP: SINGLE-SHOT NETWORK PRUNING BASED ON CONNECTION SENSITIVITY

[2] Parameter Efficient Training of Deep Convolutional Neural Networks by Dynamic Sparse Reparameterization

[3] PICKING WINNING TICKETS BEFORE TRAINING BY PRESERVING GRADIENT FLOW

[4] Rigging the Lottery: Making All Tickets Winners

[5] MEST: Accurate and Fast Memory-Economic Sparse Training Framework on the Edge


* It is not clear what the content of Fig.1 means. There are no legends for those strips.
It would make the paper easier to read if the author can explain what the “network calibration” means at the beginning of the introduction section.


* I think this is yet another paper that abuses term of “Lottery Tickets”. I really cannot find the connection between the proposed method and the Lottery Tickets hypothesis. This makes reader feel confusing.


* I don’t think it is fair to compare the proposed method with the (original) LTH method.

* The author argues that the methods using pretraining – pruning are costly. But the author does not provide comparison results about the training costs (FLOPs) of the proposed method. It is not clear the 200 training epochs is for each sub-network or for the entire training process.

* It is inappropriate to call the inference FLOPs the inference speed.

* It seems that the prior work [6] is highly related to the proposed method. But the author does not provide any discussion about it. (Only compared with it in the results part.)

[6] Calibrate and prune: Improving reliability of lottery tickets through prediction calibration.


**Questions:**

Please refer to the weakness part.

---

> ### Author Rebuttal · Authors · 2023-08-09
>
> We would like to thank the reviewer for the valuable comments/suggestions. We summarize our responses as follows.
>
>
> **Q1: Authors completely ignore very popular (static/dynamic) sparse training methods.**
>
> Please refer to the answer to Q3 of the general response. To more clearly demonstrate this critical limitation from existing sparse training methods, we conduct additional experiments to evaluate the calibration performance of all the methods suggested by the reviewer. The results are summarized in  Table 2 (refer to **the attached pdf in the general rebuttal**). It is clear that all these methods achieve an ECE score similar to another representative sparse training method EP, which is much worse than the proposed DRE framework. Therefore, our work proposes a novel contribution to achieve calibrated sparse network training that is orthogonal and complementary to existing sparse training methods.
>
>
>
> **Q2: Clarity of Figure 1.**
>
> In Figure 1, we use the standard Expected Calibration Error (ECE) plot, which is commonly used to visualize the calibration behavior of a model (see Figures 1 \& 4 in [7] as an example). In the ECE plot, the dashed diagonal line indicates that the model is perfectly calibrated where the model's accuracy exactly matches its confidence. The shaded area using the strips indicates that the model's confidence is higher than accuracy, which implies the overfitting behavior. The sky blue shows the model's accuracy for the given confidence value. If the accuracy exceeds the diagonal line, it means that the model's accuracy is higher than its confidence, which implies that the model tends to provide less confidence predictions.
>
>
>
> **Q3: Connection between the proposed method and the Lottery Tickets hypothesis.**
>
> We would like to cite several relevant works from existing literature to clearly show the connection between the proposed method and LTH. According to [8] and [9], the strong Lottery Tickets Hypothesis states that "there exists a subnetwork in a randomly initialized neural network such that it already achieves almost the same accuracy as a fully trained network, without any optimization of the weights of the network". The proposed method builds upon Edge Popup, which directly finds a sparse sub-network from a randomly initialized dense network without pre-training and iterative pruning. Therefore, it matches the definition of strong Lottery Tickets Hypothesis.
>
>
> **Q4: It is not fair to compare the proposed method with LTH method.**
>
> As a representative sparse network training method, we empirically show that LTH also suffers from poor calibration performance through our comparison. Furthermore, compared with LTH, our technique does not involve pre-training followed by iterative pruning and finetuning, making it computationally more efficient. Since we keep all other factors (e.g., data split) the same, we would like to kindly clarify that this is a fair comparison.
>
>
> **Q5: Training cost of the proposed method.**
>
> We would like to make it clear that each sparse subnetwork is trained using 200 epochs. It should be noted that there is no dependency among those sparse subnetworks (their focus on the data space is automatically adjusted by the DRO framework) and therefore can be run independently making overall training time more efficient. In contrast to our work, LTH variants (especially methods using pretraining-pruning) typically involve sequential process and therefore cannot be run parallel making them time-consuming.
>
>
> **Q6: Inappropriate to call the inference FLOPs the inference speed.**
>
>
> Thank you for the suggestion. We will fix the issue and change to inference FLOPs in the revised paper.
>
> **Q7: Discussion about prior work in [6].**
>
> Thank you for the suggestion and we will add a discussion of [6]. We would also like to clarify that [6] directly applies several commonly known calibration techniques on the top of the standard LTH model. Since it does not introduce any new technique, we only briefly discuss it when describing the performance comparison (lines 311-312).
>
>
>
> **References**
>
> - [1] SNIP: SINGLE-SHOT NETWORK PRUNING BASED ON CONNECTION SENSITIVITY
>
> - [2] Parameter Efficient Training of Deep Convolutional Neural Networks by Dynamic Sparse Reparameterization
>
> - [3] PICKING WINNING TICKETS BEFORE TRAINING BY PRESERVING GRADIENT FLOW
>
> - [4] Rigging the Lottery: Making All Tickets Winners
>
> - [5] MEST: Accurate and Fast Memory-Economic Sparse Training Framework on the Edge
>
> - [6] Calibrate and prune: Improving reliability of lottery tickets through prediction calibration.
>
> - [7] Guo et al. "On Calibration of Modern Neural Networks". ICML2017.
>
> - [8] Ramanujan et al. "What’s Hidden in a Randomly Weighted Neural Network?". CVPR2020.
>
> - [9] Chijiwa et al. "Pruning Randomly Initialized Neural Networks with Iterative Randomization". NeurIPS2021.

---

> ### Author Response · Authors · 2023-08-18
>
> Dear Reviewer RrYi,
>
> Thank you for providing your reviews and valuable suggestions!
>
> In responding to the raised concerns, we believe that the paper has been strengthened significantly and we thank the reviewer for that. The multiple sparse training methods suggested by the reviewer become very useful to further showcase the effectiveness of our proposed technique. Further, the performance gap between our technique and those existing sparse training methods as shown in our additional experimental results in our rebuttal helps to further strengthen our empirical evaluation. Also, suggestions such as using inference speed instead of FLOPs in the Table 10 would be helpful to make the paper more informative. The suggestion regarding the clarity of Figure 1 would be very helpful to make Figure 1 easier to understand.
>
> We hope that the reviewer finds our answers satisfactory and considers updating the assessment accordingly! We will be more than happy to provide any additional clarifications if needed.

---

### Official Review · Reviewer_xGvL · 2023-07-06

**Soundness:** 3 good
**Presentation:** 3 good
**Contribution:** 3 good
**Rating:** 5
**Confidence:** 3

**Summary:**

The paper utilizes Distributionally Robust Optimization (DRO) framework to achieve an ensemble of lottery subnetworks for better calibration performance. Recently developed sparse network training methods, such as Lottery Ticket Hypothesis (LTH) and its variants, largely focus on sparsifying deep networks and realizing comparable accuracy to dense counterparts but neglect network calibration. The proposed DRO ensemble aims to learn multiple diverse and complementary sparse sub-networks with the guidance of uncertainty sets, which encourage tickets to gradually capture different data distributions from easy to hard and naturally complement each other. The authors theoretically justify the strong calibration performance of their proposed robust training process and show extensive experimental results on several benchmarks, demonstrating clear calibration improvement without sacrificing accuracy or burdening inference costs. Experiments on out-of-distribution (OOD) datasets also demonstrate the robustness of their approach in the open-set environment.

**Strengths:**

The idea of using multiple subnetworks to the ensemble for better calibration performance is novel, and the procedure does not need to first train the dense network.  The authors also give a theoretical analysis of the method.  The authors conduct extensive experiments to validate the proposed method.

**Weaknesses:**

For the out-of-distribution comparison, the authors miss some important methods to compare such as  [1]. All the experiments are conducted on ResNet architectures, could the authors provide results for some other architectures? And could the proposed method be conducted with filter-level subnetworks?

[1] Zhang, Dinghuai, et al. "Can subnetwork structure be the key to out-of-distribution generalization?." International Conference on Machine Learning. PMLR, 2021.
[2] Zhou, Xiao, et al. "Sparse invariant risk minimization." International Conference on Machine Learning. PMLR, 2022.

**Questions:**

Please refer to Weakness.

**Limitations:**

Yes.

---

> ### Author Rebuttal · Authors · 2023-08-09
>
> We would like to thank the reviewer for the valuable comments/suggestions. We summarize our responses as follows.
>
> **Q1: Comparison with OOD Works [1, 2].**
>
> Thank you for pointing out these important references, which we would like to include and discuss in the related works section of the revised paper. Here, we would like to highlight some key differences from our work.
>
> - First, the training procedure of [1] also follows the dense pretraining  paradigm, where dense network pretraining is performed followed by the pruning and finetuning of the pruned network. This process is similar to the LTH work, which we have already included as a comparison baseline. As we can see from Table 2, dense pretraining techniques tend to perform poorly on OOD data leading to a lower ECE score. The underlying reason for the poor calibration on OOD data is associated with the dense network pretraining and is also explained by [2]. Through dense network training, the model may have already picked the spurious and noisy features while paying less attention to the important ones. As such, during the pruning phase, the subnetwork is searched by giving preference to the spurious and noisy features, leading to a suboptimal calibration performance. Compared to this, our proposed DRE does not rely on the pretrained dense network. The chance of finding sparse subnetworks that capture important features is also enhanced thanks to the DRO based training process.
>
>
> - In the case of [2], the authors propose Sparse Invariant Risk Minimization (SIRM) that applies sparse techniques on the top of IRM to extract the invariant features in the desired sparse subnetwork. This paper demonstrates that IRM alone is not enough to avoid spurious features and therefore sparsity is required during the training process to learn from the invariant features. As such, the resulting subnetwork will have an enhanced generalization capability.  Notably, the framework in [2] can not be compared directly with our DRE technique as IRM training requires datasets from multiple environments. Meanwhile, SIRM is a generic technique that can be integrated with any desired sparse training strategies. Therefore, SIRM is complementary and can be used together with our DRE framework.
>
>
>
> **Q2: All the experiments are conducted on ResNet architectures, could the authors provide results for some other architectures?**
>
> Thank you for raising this great point! In fact, we have conducted an ablation study that includes investigates the impact of different architectures, and the results are summarized in Table 9 of Appendix D.8.  Using a vision transformer (i.e., ViT), DRE still achieves a much lower calibration error than EP. However, using ViT as a backbone, the accuracy from both EP and DRE is lower and ECE is higher than other backbones. It has been shown by existing works that without pretraining, the lack of useful inductive biases for ViT can cause a performance drop. Since no pretraining is conducted in both EP and DRE, it causes a lower accuracy (and a higher ECE).
>
>
> **Q3: Proposed method on filter-level subnetworks.**
>
> Thank you for this thoughtful comment! In our experimentation, backbones contain convolution layers and we have explicitly enforced sparsity in each convolution module. If we understand this question correctly, we believe filter-level subnetworks indicate the sparse subnetworks and if so, we have already considered filter-level sparsity in our framework. One of the closest works that analyze the implicit sparsity in the filter level is [3]. In this paper, the authors empirically show the sparsification of convolution neural networks mainly with respect to (a) Regularizer (weight decay and $L_2$),  (b) Optimizer (SGD, Adam, Adadelta, and Adagrad), and (c) Difficulty of the task. They found that the adaptive optimizers (e.g., Adam, Adadelta, and Adagrad) learn sparser network representation compared to SGD. Also, no sparsity is observed in the absence of regularizers like $L_2$ and weight decay. Also, the sparsity depends on the interplay between the regularizer and optimizer. For example, $L_2$ shows a higher sparsity for comparable performance than weight decay when using the Adam optimizer, whereas using SGD the difference is not significant using weight decay and $L_2$. Furthermore, they showed that it is difficult to obtain a sparser network for difficult tasks whereas higher sparsity can be more easily obtained using less difficult tasks (e.g., Cifar10 generates higher sparsity compared to Cifar100). However, the analysis does not seem to align with our goal, which aims to improve the calibration ability of sparse networks whereas, [3] primarily focuses on analyzing the implicit sparsity achieved by considering different factors. We have conducted an experimentation considering different optimizers and weight decay coefficient in the Cifar10 dataset with ResNet50 architecture As shown in Table 4 (a) (in **the attached pdf file of the general rebuttal**), simply using the different optimizers with varied weight decay coefficients does not help to improve the calibration.
>
> **References**
>
> - [1] Zhang, Dinghuai, et al. "Can subnetwork structure be the key to out-of-distribution generalization?." International Conference on Machine Learning. PMLR, 2021.
>
> - [2] Zhou, Xiao, et al. "Sparse invariant risk minimization." International Conference on Machine Learning. PMLR, 2022.
>
>
> - [3] Mehta et al. "On Implicit Filter Level Sparsity in Convolutional Neural Networks". CVPR2019.

---

> > ### Comment · Reviewer_xGvL · 2023-08-21
> >
> > Thanks for the detailed rebuttals. I will keep my score.

---

> ### Author Response · Authors · 2023-08-18
>
> Dear Reviewer xGVL,
>
> Thank you for providing your reviews and valuable suggestions!
>
> By addressing raised concerns, we believe that the paper has been improved and we thank the reviewer for that. The references related to out-of-distribution works will help us further strengthen our motivation and would be helpful to justify choosing the sparse sub-network technique instead of the dense network in our framework. Also, a comparison with filter-level subnetworks in our answer to Q3 becomes helpful to further demonstrate that a simple combination of regularization techniques (such as weight decay) and optimizers is inadequate to improve the calibration performance.
>
> We hope that the reviewer finds our answers satisfactory and considers updating the assessment accordingly! We will be more than happy to provide any additional clarifications if needed.

---

### Official Review · Reviewer_TYXu · 2023-07-07

**Soundness:** 2 fair
**Presentation:** 3 good
**Contribution:** 2 fair
**Rating:** 5
**Confidence:** 3

**Summary:**

The authors propose a method of sparse training of deep neural networks with the objective of confidence calibration. The method is based on learning an ensemble of sparse models where each model begins with the same training dataset, and is increasingly diversified such that each model in the ensemble is trained on different sets of hard/rare data samples. The authors walk through a theoretical argument on why they believe this helps to improve confidence calibration based on prior work on understanding spurious correlations and training. Results are demonstrated on Res/Nets with CIFAR-10, CIFAR-100 and Tiny ImageNet for ensemble models of 91% and 85% sparsity (considered over all models in ensemble). Results are compared with sparse training baselines, such as L1 pruning, CigL, Sup-Ticket and DST Ensemble, along with Sparse Network Ensemble and finally a single dense model.

**Strengths:**

- The authors focus on confidence calibration, and evaluate their results not just in a typical classification context, but also in out-of-distribution and open-set distributions where the effect of confidence calibration is well-motivated and clear.
- Results are on reasonable models and datasets, with mostly appropriate baselines (at least on the sparsity side)
- Results suggest compared to dense and most sparse baselines the proposed method improves confidence calibration significantly.
- The proposed "Robust loss" which focuses on ensembles learning from a diverse set of difficult examples found during training is intuitive.
- Reasonable presentation overall, although could be cleaned up and made easier to understand.

**Weaknesses:**

- I found the motivation of using sparse ensembles (v.s. a dense ensemble) lacking, with no clear reason why the proposed method would not work for a dense ensemble also, and notably the lack of a dense ensemble as a baseline. If the motivation is improved training performance, there is no description or qualification of the improved performance, with a comparison of the dense baseline to understand any potential tradeoffs, if the motivation is improved calibration compared to dense ensemble, then similarly this should be also be explicitly shown.
- No ablation analysis of different effects of various aspects of the proposed method, i.e. sparse ensemble v.s. dense ensemble v.s. proposed robust loss, in particular sparse ensemble v.s. robust loss being the most important.
- I found the "Theoretical Analysis" to be quite far from theoretical, and more motivation for the proposed method than a proof of any sort. Indeed considering how poorly spurious correlations are understood in the field, I believe the author's apparent claims that (a) the over-confidence of neural networks is due to spurious correlations alone, and (b) the approach proposed by the authors addresses spurious correlations can only be an over-claim. While this section is fine as a motivation for the method, this cannot be claimed to be a proof or formal demonstration that the method reduces spurious correlations in general.
- What appears to be one of the most important/strongest baselines (DST Ensembles) is missing from all but the CIFAR-10 and CIFAR-100 classification results.
- For TinyImageNet a sparse ensemble without the training data curriculum appears to have a relatively similar effect at the highest sparsity level as the proposed method on improving confidence calibration drastically, again causing me to wonder about ablation analysis of the method.

Minor issues:
- Writing is fairly verbose and in some cases a bit repetitive, could be made more easy to understand/compact.
- Confusing usage of "density" rather than "sparsity" unlike most of the literature in the field.

**Questions:**

- What is the motivation of using sparse v.s. dense ensembles in the proposed method?
- What is the effect of the different aspects of the proposed method? i.e. robust loss, sparse v.s. dense training, and ensembles.
- Why are dense ensembles not a baseline?

**Limitations:**

The authors did not AFAIK address any such limitations, although improving confidence calibration of neural networks would have positive societal impact overall I believe.

---

> ### Author Rebuttal · Authors · 2023-08-09
>
> We would like to thank the reviewer for the valuable comments/suggestions. We summarize our responses as follows.
>
>
> **Q1: Motivation of using sparse ensemble.**
>
>
> Please refer to our answer to Q1 in the general response. In addition to a better calibration performance, sparse ensembles also achieve a significant computational advantage when compared with dense ensembles. Specifically, using DRE we only use 15\% of the weights present in a single dense network leading to a significant reduction in the computational cost. As shown in Table 10 of the paper, using a dense ensemble with 3 dense networks, we will have a total of 12.42 ($\times 10^{9}$) FLOPs whereas DRE only uses 1.31 ($\times 10^{9}$).
>
>
>
>
>
>
> **Q2:  Ablation study on sparse ensemble vs deep ensemble.**
>
> Thank you for this suggestion. In our response to Q1, we have compared the proposed sparse ensemble DRE with (1) a standard dense ensemble with each dense network trained using the ERM loss and (2) a dense ensemble trained using the proposed robust loss. The results ((Table 1 in attached file)) confirm that DRE is able to achieve a much better calibration performance than dense ensembles. We explain the reason for the performance difference in our answer to Q1 as well.
>
>
>
>
> **Q3: Importance of theoretical Analysis.**
>
> Please refer to our answer to Q2 in the general response. To add more empirical evidence that supports our theoretical contribution, we have conducted additional experiments on the WaterBird dataset that contains explicit spurious correlations [7]. As shown in Table 4 (b) (in **the attached pdf file of the general rebuttal**), the proposed DRE greatly improves the calibration performance compared to the sparse network ensemble (SNE) in testing samples, where there is no spurious correlations (last two columns) between the background and the bird in the image. These are the samples with waterbirds in a land background or landbirds in a water background. The result justifies that our model indeed learns from important features instead of spurious correlations. In contrast, in the case of the testing samples holding spurious correlations, i.e., waterbird in a water background and landbird in a land background (Spurious columns in Table 4 (b)), DRE and SNE achieve a comparable performance. This is because overconfident predictions are preferred on these testing samples as they are most likely to be correct benefiting from the spurious correlation.
>
>
> **Q4. DST Ensemble Baseline.**
>
> Thank you for the comment. We would like to clarify that we have included the comparison with the DST ensemble for Cifar10, Cifar100, as well as TinyImageNet. For the classification task, the
>  DST ensemble result for TinyImageNet is reported  in Table 6 of the Appendix. Furthermore, in the case of the out-of-distribution experiments on Cifar10 and Cifar100, the reason for not including the DST ensemble method is provided in lines 342-344. Specifically, in the case of the Cifar100-C, the accuracy using DST is more than **11\%** lower than that of the DRE in both architectures whereas in the case of Cifar10-C, the accuracy using DST is more than **6\%** lower than that of the DRE using both architectures.
>
>  **Q5. For TinyImageNet a sparse ensemble without the training data curriculum appears to have a relatively similar effect at the highest sparsity level as the proposed method on improving confidence calibration drastically...**
>
>  Thank you for pointing out this important observation. First of all, this provides another important empirical evidence that, compared to the dense networks, sparse models and the ensemble thereof (i.e., SNE) help to improve calibration. Second, while SNE achieves significant improvement as compared with other baselines, the proposed DRE further outperforms SNE to a large extent in most cases. For example, on the WideResNet101 backbone, the ECE score of SNE is still two times of DRE; for the sparsity level at 15\%, the ECE scores of SNE are 5-6 larger than those of DRE. The results are consistent over all datasets on all types of backbones.
>
>
>
> **Q6. The authors did not AFAIK address any such limitations...**
>
> We apologize for the confusion regarding limitations. In Appendix E.2, we have discussed the limitations and potential future extensions of our work.

---

> > ### Comment · Reviewer_TYXu · 2023-08-16
> >
> > First, I’d like to thank the authors for their rebuttal comments.
> >
> > > Q1: Motivation of using sparse ensemble.
> >
> > I’d like to clarify that I’m a fan of sparsity in general, but for any method/paper it is important to motivate the work. I believe the author’s comments here do a *much* better job at doing this, but I want to clarify if the authors have revised the paper to motivate sparse ensembles?
> >
> > > Q2: Ablation study on sparse ensemble vs deep ensemble.
> > > Thank you for this suggestion. In our response to Q1, we have compared the proposed sparse ensemble DRE with (1) a standard dense ensemble with each dense network trained using the ERM loss and (2) a dense ensemble trained using the proposed robust loss. The results ((Table 1 in attached file)) confirm that DRE is able to achieve a much better calibration performance than dense ensembles. We explain the reason for the performance difference in our answer to Q1 as well.
> >
> > Thanks for the extra experimental results/requested baseline. I think these are important results in motivating your analysis of, and proposed method in, improving the calibration using sparse ensembles, and it strengthens your results considerably.
> >
> > > Q3: Importance of theoretical Analysis.
> > > Please refer to our answer to Q2 in the general response. To add more empirical evidence that supports our theoretical contribution…
> >
> > Unfortunately this rebuttal answer misses my point entirely, while simultaneously emphasizing it by discussing empirical results only: *there is no theoretical analysis present in your paper*. I’m a big fan of using empirical results to support a hypothesis, and I’d personally be happy if that’s what your paper claimed vis-a-vis spurious examples and your method, but that is not what your paper claims to be doing, or you claim in this rebuttal. Instead you appear to be continuing to claim you have a theoretical analysis, and this is simply not true. It is not acceptable to publish a paper claiming to have a theoretical analysis/support for a claim when it does not.
> >
> > > Q5. For TinyImageNet a sparse ensemble without the training data curriculum appears to have a relatively similar effect at the highest sparsity level as the proposed method on improving confidence calibration drastically...
> > > …..  The results are consistent over all datasets on all types of backbones.
> >
> > My point here is that this result isn’t consistent with this story/the other results, so I think it would be good to explain it, especially since this model/dataset doesn’t seem to benefit much from the proposed method.
> >
> > *Summary*
> >
> > While I found some of the author’s rebuttal comments and the extra experiments could address many of my issues with the work, especially in lack of motivation and baselines, the author’s continued insistence that they have a theoretical analysis supporting their method and hypothesis as to the mechanism behind that method — when they simply don’t — is not acceptable. Perhaps more frustratingly, I see little need for the over-claims or anything more than empirical analysis to support their method, and would advise the authors to revaluate how they present this work.

---

> > > ### Author Response · Authors · 2023-08-17
> > >
> > > Many thanks for carefully checking our rebuttal and providing additional comments and valuable follow-up questions. We appreciate the confirmation from the reviewer that our rebuttal along with the extra experiments could address many concerns raised in the original review, especially in lack of motivation and baselines. In what follows, we provide our response to each of the follow-up questions.
> > >
> > > **...if the authors have revised the paper to motivate sparse ensembles?**
> > >
> > > We would like to mention that during the rebuttal phase we are not allowed to make changes in the paper. We will incorporate all the changes that we made during the rebuttal phase in the revised paper.
> > >
> > > **Theoretical analysis**
> > >
> > > Regarding the theoretical analysis, sorry for misinterpreting your comment in our first response and thank you for your further clarification! We would like to acknowledge that our analysis in Section 3.3 is primarily from the spurious correlation perspective so it is by no means a complete and thorough theoretical analysis to fully understand the over-confidence issue present in the deep neural networks. Our goal is trying to offer some deeper insight on why the proposed approach may work so that the good performance could be potentially generalized to a wider range of datasets and network architectures. The additional experiments conducted in our rebuttal offers more concrete evidence to support the conclusions made in that section. To this end, we formalize the problem setup by clearly stating the assumptions and conduct some formal mathematical derivations to arrive at our conclusions. It is worth noting that our analysis also builds upon some recent theoretical advances (such as Zeyuan Allen-Zhu and Yuanzhi Li. Towards understanding ensemble, knowledge distillation and self-distillation in deep learning, ICLR 2023) but makes novel adaptions to our unique problem setting.
> > >
> > > We would appreciate if the reviewer could provide some additional guidance on how to improve this part. If our analysis lacks rigorousness that does not meet the standard of a typical theoretical analysis, we could tone down any statements on the theoretical contribution.
> > >
> > > **Result on TinyImageNet**
> > >
> > > Regarding the TinyImageNet result, thank you again for further clarifying the question. This is a great point! We believe the reason for the smaller gap between SNE and DRE in the case of TinyImageNet may be  related to the overfitting phenomenon. In the case of TinyImageNet, because of the difficulty of the dataset, we would need a relatively larger network architecture compared to the Cifar10 and Cifar100 in order to capture the complex patterns in the data. As such, sparse subnetworks constructed from small to mid sized networks are less likely to overfit (or even underfit). Therefore, the SNE ensemble model will also be relatively less overfitted compared to Cifar10 and Cifar100. It is also interesting to observe that as we move to a larger backbone (i.e., from ResNet101 to WideResNet101), the gap between SNE and DRE increases (along with the chance of overfitting). Furthermore, with a higher density at 15\%, the gap between SNE and DRE becomes even larger, which is consistent with our analysis above.
> > >
> > > Thanks again for your feedback and we are happy to answer any follow-up questions.

---

### Author Rebuttal · Authors · 2023-08-09

First of all, we would like to thank all the reviewers for spending time to review our paper and providing many constructive suggestions and comments. Here, we summarize our responses to some major questions raised by reviewers:

**Q1: Motivation of using sparse ensemble (Reviewer TYXu)**

Besides a high computational cost, one key motivation for developing sparse ensembles is the poor calibration of dense networks resulting from the memorization effect introduced by an over parameterized architecture as mentioned at the beginning of the introduction section. Such phenomenon has been commonly observed and discussed in recent literature (e.g., refs [9, 24] in the paper along with some other references such as [1] and [2] suggested by other reviewers). Our empirical study has further confirmed this across different datasets and backbone architectures. For example, ref [24] theoretically justifies that overparameterization in dense networks exacerbates spurious correlation leading to poor calibration; Refs [1] and [2] (see references for details)  further demonstrate that dense networks are more likely to overfit leading to poor generalization (and calibration). To avoid  overfitting,  [1] and [2] also resort to training sparse networks. Our empirical evaluation shows that sparse network training can consistently improve the calibration performance over multiple datasets in multiple architectures. As can be seen in Tables 1-3 of the paper, sparse networks achieve better ECE scores than their dense counterparts in most settings.

To more clearly demonstrate the advantage of using the proposed DRE compared to a dense ensemble, we have conducted additional experimentation and present the results in Tables 1 presented in **the attached pdf file**. The Dense Ensemble (w/o DRO) refers to the one where we ensemble multiple dense networks, where each one is trained using the standard ERM loss. The Dense Ensemble (w/ DRO) is the one where we train multiple dense networks but using the DRO loss (i.e., Eq. 1). As can be seen, the proposed sparse ensemble (i.e., DRE) clearly outperforms the dense ensembles to a large extent. It is also interesting to note that a dense ensemble (with DRO) only achieves a slightly better ECE score as compared with a dense ensemble (w/o DRO). This is because, it is more difficult to further diversify dense networks with the exactly same architectures (i.e., nodes and connections). In contrast, using sparse training, we can naturally pick very distinct sparse subnetworks from the original dense network to increase the diversity, where each subnetwork is already better calibrated because of the reason explained above. Additionally, thanks to the distributionally robust ensemble, we can further diversify the learned subnetworks leading to much better calibration performance.

**Q2: Importance of theoretical Analysis. (Reviewer TYXu)**

We agree with the reviewer that our theoretical analysis can further strengthen our motivation. We also acknowledge that spurious correlation is only one potential source that can lead to over-confidence in neural networks (but we did not claim that *over-confidence of neural networks is due to spurious correlations alone*). Our analysis primarily focuses on this important perspective, which identifies concrete theoretical evidence on this important connection. The theoretical findings have been further confirmed in our empirical evaluations on multiple datasets over a diverse set of backbone architectures. Therefore, we believe our work is an important step forward in addressing this highly important issue by providing a theoretically sound and empirically feasible solution to effectively lower the model’s false confidence on its wrong predictions resulting from spurious correlations, as evidenced by the improved overall calibration performance. The analysis of other sources that contribute to overconfident predictions remains as an important topic for future research.

**Q3: Authors completely ignore very popular (static/dynamic) sparse training methods. (Reviewer RrYi)**

Thank you for pointing out these sparse training methods without the need for iterative pruning/growing. To most clearly differentiate these works from our main technical contribution, we would like to re-state our primary focus, which is to **achieve calibrated sparse network training**, as indicated by the title of our paper. To this end, we propose a novel Distributionally Robust Optimization (DRO)
 framework to achieve an ensemble of lottery tickets towards calibrated network sparsification (see lines 8-10 of the abstract). Like the reviewer, we clearly recognize the existence of sparse training techniques that do not rely on pre-training and iterative pruning (see lines 33-34). We use Edge Popup (EP) as a representative of this group of methods (and the plural implies that EP is only one of such methods). However, like EP, all these methods focus on pushing the accuracy up to the original dense networks and hence still suffer from a severely overfitting behavior, leading to a poor calibration performance. This is the *novel technical gap* that we identify and aim to address using our proposed Distributionally Robust Ensemble (DRE). We have empirically shown the poor calibration performance from EP (along with some other representative sparse training models, including [3] as mentioned by the reviewer) in both Figure 1 and our experimental results in Tables 1-3 in the main paper and Tables 5-8 in the Appendix.


**References**

- [1] Zhang, Dinghuai, et al. "Can subnetwork structure be the key to out-of-distribution generalization?." International Conference on Machine Learning, 2021.

- [2] Zhou, Xiao, et al. "Sparse invariant risk minimization." International Conference on Machine Learning, 2022.

- [3] Evci, U., et al. Rigging the lottery: Making all tickets winners. In International Conference on Machine Learning, 2020.

---

> ### Comment · Reviewer_TYXu · 2023-08-16
>
> As this is a general comment by the authors, I’d like to clarify for the sake of other reviewers/the AC that “Q2” here completely misrepresents what I stated about the author’s “theoretical motivation:
>
> > Q2: Importance of theoretical Analysis. (Reviewer TYXu)
> > We agree with the reviewer that our theoretical analysis can further strengthen our motivation. …
>
> In fact I stated that the “theoretical analysis” could only at best be called motivation, as it doesn’t present any theoretical analysis that I can see.

---

> > ### Author Response · Authors · 2023-08-17
> >
> > Dear Reviewer TYXu,
> >
> > Sorry for misinterpreting your comment in our first response and thank you for your further clarification! We would like to acknowledge that our analysis in Section 3.3 is primarily from the spurious correlation perspective so it is by no means a complete and thorough theoretical analysis to fully understand the over-confidence issue present in the deep neural networks. **Our goal  is trying to offer some deeper insight on why the proposed approach may work so that the good performance could be potentially generalized to a wider range of datasets and network architectures. The additional experiments conducted in our rebuttal offers more concrete evidence to support the conclusions made in that section.** To this end, we formalize the problem setup by clearly stating the assumptions and conduct some formal mathematical derivations to arrive at our conclusions. It is worth noting that our analysis also builds upon some recent theoretical advances (such as Zeyuan Allen-Zhu and Yuanzhi Li. Towards understanding ensemble, knowledge distillation and self-distillation in deep learning, ICLR 2023) but makes novel adaptions to our unique problem setting.
> >
> > We would appreciate if the reviewer could provide some additional guidance on how to improve this part. If our analysis lacks rigorousness that does not meet the standard of a typical theoretical analysis, we could tone down any statements on the theoretical contribution.
> >
> > Best,
> >
> > Authors

---

### Decision · Program_Chairs · 2023-09-21

**Decision:**

Accept (poster)

**Comment:**

The paper received 4 reviews, and the authors did a solid job in addressing all the reviewer comments and suggestions. This includes running new experiments to analyze performance and including more comparative methods. After the rebuttal and the discussion period, three reviewers were leaning to accept the paper. The rebuttal has addressed the majority of the reviewer concerns and the remaining issues seem to be minor.

Reviewer TYXu raised a fair concern about the theoretical analysis and the claims being made based on it. This reviewer decided to raise their score after the rebuttal/discussion phase under the assumption that the authors phrase their claims on the theoretical analysis (as the rebuttal suggests) to be more realistic as to what is being shown in the paper and how it relates to the proposed method. This is a fair request and the authors are encouraged to address this in their final version.